# Entropy-stabilized single-atom Pd catalysts via high-entropy fluorite oxide supports

Haidi Xu [1,2,3], Zihao Zhang[2,3], Jixing Liu[3], Chi-Linh Do-Thanh [3], Hao Chen[3], Shuhao Xu[4], Qinjing Lin[4], Yi Jiao[1], Jianli Wang[4], Yun Wang [5✉], Yaoqiang Chen [1,4✉] & Sheng Dai [2,3✉]

Single-atom catalysts (SACs) have attracted considerable attention in the catalysis community. However, fabricating intrinsically stable SACs on traditional supports (N-doped carbon, metal oxides, etc.) remains a formidable challenge, especially under high-temperature conditions. Here, we report a novel entropy-driven strategy to stabilize Pd single-atom on the high-entropy fluorite oxides $(CeZrHfTiLa)O_x$ (HEFO) as the support by a combination of mechanical milling with calcination at 900 °C. Characterization results reveal that single Pd atoms are incorporated into HEFO ($Pd_1$@HEFO) sublattice by forming stable Pd–O–M bonds (M = Ce/Zr/La). Compared to the traditional support stabilized catalysts such as Pd@$CeO_2$, $Pd_1$@HEFO affords the improved reducibility of lattice oxygen and the existence of stable Pd–O–M species, thus exhibiting not only higher low-temperature CO oxidation activity but also outstanding resistance to thermal and hydrothermal degradation. This work therefore exemplifies the superiority of high-entropy materials for the preparation of SACs.

[1] Institute of New Energy and Low-Carbon Technology, Sichuan University, Chengdu 610064, China. [2] Chemical Sciences Division, Oak Ridge National Laboratory, Oak Ridge, TN 37831, USA. [3] Department of Chemistry, Joint Institute for Advanced Materials, University of Tennessee, Knoxville, TN 37996, USA. [4] College of Chemistry, Sichuan University, Chengdu 610064, China. [5] Sinocat Environmental Technology Co. Ltd., Chengdu 611731, China. ✉email: wangy@sinocat.com.cn; nic7501@scu.edu.cn; dais@ornl.gov

igh-entropy alloys (HEAs) with five or more elemental species have been successfully synthesized and have attracted extensive attention due to their unique physical properties and potential applications[1–3]. Compared with HEAs, it is more difficult to yield the formation of high entropy ceramics including oxides, carbides, nitrides et al. owing to their larger heats of formation. Since 2015, the family of high entropy materials has been expanded when the first entropy-stabilized oxides (HEOs) were reported by Rost et al.[4]. HEOs, a kind of multicationic equiatomic oxides, were then found to have a variety of interesting and unexpected characteristics[5]. Inspired from this discovery, several kinds of high-entropy oxides with different crystal structures were then reported one after another, such as perovskite oxide[6,7], spinel oxide[8], and fluorite oxide[9–11]. Quite recently, new entropy-stabilized single-phase fluorite oxides (HEFO) $Hf_{0.25}Zr_{0.25}Ce_{0.25}Y_{0.25}O_{2-\delta}$[11] and $Ce_{0.2}Zr_{0.2}Hf_{0.2}Sn_{0.2}Ti_{0.2}O_2$[12] were also synthesized by using high-energy ball milling. However, rigorous synthetic conditions, including the long reaction time (6–24 h), ultra-high temperature (1500–1800 °C), and uniaxial high pressure, limit its application in catalysis.

Fluorite oxide-supported noble metals including SACs are widely used as heterogeneous catalysts with superior catalytic performances for mitigating critical pollutants (e.g., CO, $CH_4$, HCs, and $NO_x$) from engine emissions. A wide range of hosts (metal oxides, N-doped C, zeolites, MOFs, etc.) are known for their ability to stabilize single atoms[13]. Recent advances in synthetic strategies for SACs are generally dominated by wet-chemistry approaches including defect engineering, spatial confinement, and coordination design strategies[14,15]. Mechanochemistry scenarios, which have attracted more interest because of their quick, quantitative, and solvent-free properties compared with wet-chemistry approaches, however, have always been a great challenge for fabricating atomically dispersed metal sites[16–18]. In addition, the successful assembly of metal single atoms on conventional carriers often requires a very careful control of synthesis conditions, such as low temperature[19,20], low metal loading[21,22], grafting of N-rich organic linkers[23,24], etc. Therefore, fabricating sintering-resistant SACs with intrinsically thermodynamic stability on high-entropy supports by using a solvent-free synthetic strategy is highly desirable.

Herein, we present a mechanochemical-assisted synthesis of Pd single atoms substituted on HEFO by simple mechanical milling followed by the calcination of metal precursors. The unique trait of HEFO can be validated by the formation of isolated Pd atoms on $Pd_1@HEFO$, as well as the presence of Pd agglomeration with $CeO_2$ as an alternative carrier under the same preparation conditions. A series of characterizations including high-resolution transmission electron microscopy (HRTEM), energy-dispersive X-ray spectroscopy (EDS) mapping, extended X-ray absorption fine structure (EXAFS), and diffuse reflectance infrared Fourier transform spectroscopy (DRIFTS) measurements are employed to unravel the existence of high-entropy phase and isolated Pd atoms in $Pd_1@HEFO$. The catalytic activity of CO oxidation, as well as the resistance to thermal and hydrothermal degradation are then compared for $Pd_1@HEFO$ and $Pd@CeO_2$ catalysts to prove the advantages of HEFO as the catalyst carrier.

## Results

### Synthesis and characterizations
As illustrated in Fig. 1, six metal salt precursors (Ce, Zr, Hf, Ti, La, and Pd) are first mixed with fumed silica by ball milling. The resultant mixture is pyrolyzed at 900 °C in air to achieve the silica-templated metal oxide complex. Finally, $Pd_1@HEFO$ catalyst is obtained after etching silica with NaOH. It should be noted that the molar ratios of Ce, Zr, Hf, Ti, and La are approximately 1:1:1:1:1 confirmed by both EDS and

inductively coupled plasma (ICP) results listed in Supplementary Table 1. The high surface-area HEFO is synthesized through the same steps without the addition of Pd precursor; $Pd@CeO_2$ counterpart is prepared by the same method; the single-phase $CeO_2$, $TiO_2$, $ZrO_2$, $La_2O_3$, and $HfO_2$ are obtained from facile pyrolysis of their corresponding metal salts at 900 °C.

Powder X-ray diffraction (PXRD) (Fig. 2a) is performed to demonstrate the crystalline structure of HEFO and its corresponding single metal oxides. The cubic $CeO_2$ ($c$-$CeO_2$), monoclinic $ZrO_2$ ($m$-$ZrO_2$), monoclinic $HfO_2$ ($m$-$HfO_2$), tetragonal $TiO_2$ ($t$-$TiO_2$), and tetragonal $La_2O_3$ ($t$-$La_2O_3$) are observed for the single metal oxides, respectively. Interestingly, HEFO exhibits only five obvious broad peaks centered at 30.2, 34.8, 50.2, 60.1, and 62.6°, corresponding to (111), (200), (220), (311), and (222) planes of a single cubic phase. The absence of diffraction peaks indexed to $m$-$ZrO_2$, $m$-$HfO_2$, $t$-$TiO_2$, and $t$-$La_2O_3$ indicates that Zr, Hf, Ti, and La are all incorporated into $c$-$CeO_2$ to form a new high-entropy $(CeZrHfTiLa)O_x$ solid solution (HEFO). Moreover, $Zr^{4+}$, $Hf^{4+}$, and $Ti^{4+}$ except for $La^{3+}$ have a smaller ion radius than $Ce^{4+}$, thus resulting in an obvious shift of diffraction peaks of HEFO to a higher $2\theta$ value compared with $c$-$CeO_2$. The structural and chemical uniformity of HEFO is further evidenced by high-resolution TEM (HRTEM) and fast Fourier transfer (FFT, Fig. 2j) images, which show well-defined lattice fringes without the secondary phases. In addition, EDS-mapping results show the highly homogeneous dispersion of randomly-distributed five metal signals including Ce, Zr, Hf, Ti, and La (Fig. 2c–i), which also unambiguously suggests the formation of the high-entropy cubic phase of HEFO on nanometer scale. $N_2$ adsorption–desorption isotherm and corresponding pore size distribution (Supplementary Figs. 1 and 2) of HEFO exhibit the emergency of rich porosity due to removal of hard-template $SiO_2$ with a high specific surface area of 162.1 $m^2 g^{-1}$ (Supplementary Table 2). This porous structure of HEFO makes it a suitable candidate to be a catalyst carrier. The surface components of HEFO is mainly dominated by $Zr^{4+}$, $Hf^{4+}$, $Ti^{4+}$, $Ce^{4+}$, and $La^{3+}$ shown by X-ray photoelectron spectroscopy (XPS, Supplementary Fig. 3) analysis, which indicates partial removal of oxygen in $(CeZrHfTiLa)O_x$ ($x < 2$) crystal after incorporation of La compared with $c$-$CeO_2$. The schematic model of cubic HEFO is then constructed (Fig. 2k) based on the above results, where the Ce atoms in $c$-$CeO_2$ is randomly populated by Zr, Hf, Ti, and La atoms.

After introducing 0.5–2 wt% Pd during the synthesis of HEFO, the HEFO crystalline phase is well retained without the appearance of any additional diffraction peaks in Fig. 2b. More importantly, the absence of diffraction peaks ascribed to Pd species and the shift of diffraction peaks with the increases of Pd loading (Fig. 2b) suggests that a large portion of Pd may be incorporated into the HEFO sublattice for the formation of $(Pd_yCeZrHfTiLa)O_x$ solid solution. After the introduction of Pd with different weight loading, the surface area and pore volume of $Pd_1@HEFO$-$x$ slightly decrease compared with pristine HEFO carrier (Supplementary Table 2). Fortunately, the surface area, pore size distribution and crystalline structure of $Pd_1@HEFO$ stay almost unchanged after both thermal and hydrothermal treatment (Supplementary Figs. 1, 2, and 4). The EDS-mapping results of $Pd_1@HEFO$ in Fig. 3a–h suggest the uniform element distribution of Pd, Ce, Zr, Hf, Ti, and La. More importantly, the agglomeration and sintering of Pd species are not observed in $Pd_1@HEFO$ (Fig. 3b), indicating the possible existence of isolated Pd sites. The HRTEM image in Fig. 3i only depicts randomly oriented lattice spacing belonging to the HEFO phase. Consistent with EDS-mapping and HRTEM results, FFT pattern in Fig. 3i (inset) again reveals the diffraction rings from (111), (200), (220), (311), and (222), attributed to the face-centered cubic

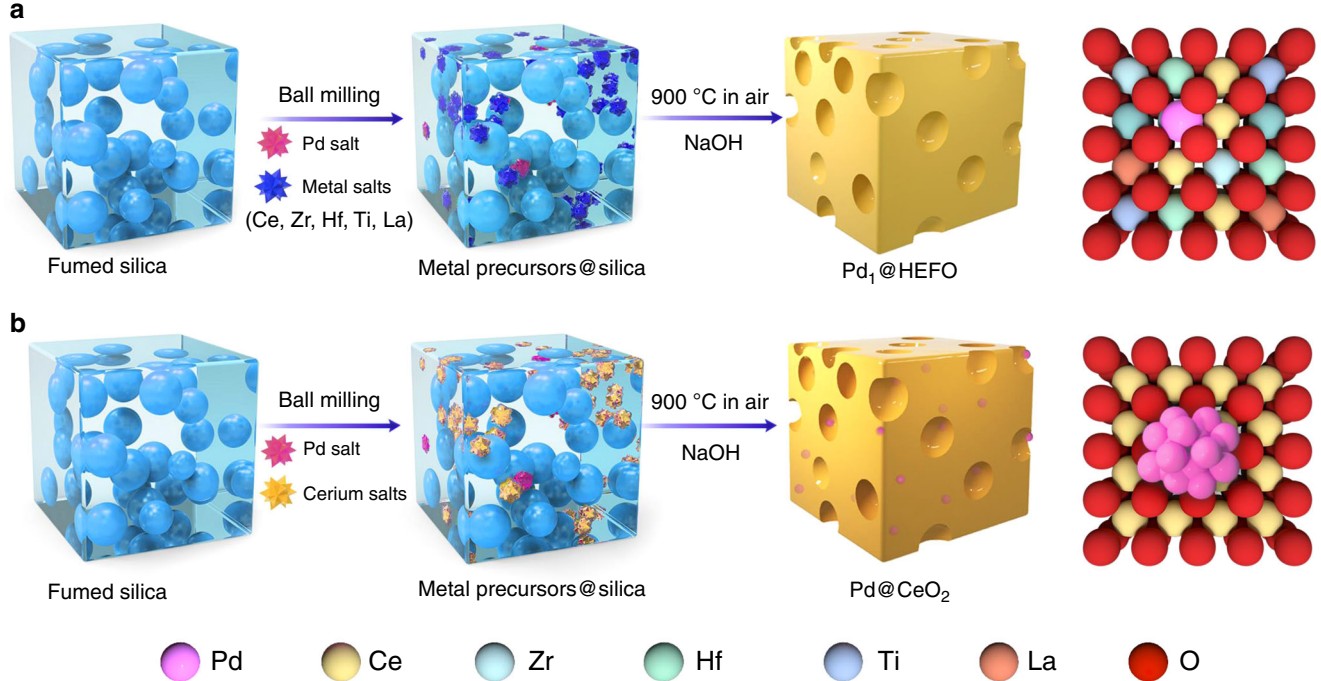

**Fig. 1 Schematic illustration of mechanochemical-assisted route for the synthesis of serial catalysts. a** Pd$_1$@HEFO, showing the possible high dispersion of a small portion of Pd single atoms on the surface of HEFO support and incorporation of a large portion of Pd single atoms into HEFO sublattice after calcination at 900 °C. **b** Pd@CeO$_2$, showing the aggregation of Pd clusters predominantly located on the surface of CeO$_2$.

(CeZrHfTiLa)O$_x$ solid solution structure without diffraction rings ascribed to any Pd species. The above results confirm the existence of isolated Pd atoms, but the microenvironment of Pd still requires further exploration. In addition, TEM images of HEFO and Pd$_1$@HEFO (Supplementary Fig. 5) display similar morphologies and the grains size distributions for both are centered at around 4 nm. This suggests that the morphology and particle size of HEFO will not change after the introduction of Pd. Furthermore, the chemical state of Pd in Pd$_1$@HEFO is then investigated by XPS analysis, and the obtained binding energy (Supplementary Fig. 6) is the characteristic of electron-deficient Pd$^{4+}$ [25–27]. This formation of electron-deficient Pd$^{4+}$ may be attributed to the electron transfer from Pd to M through Pd–O–M bonds (M = Ce, Zr, Hf, Ti, and La) in (Pd$_y$CeZrHf-TiLa)O$_x$ solid solution.

The formation of single Pd atoms is further confirmed by atomic-resolution TEM image (Fig. 4a) and the corresponding Pd EDS-mapping image (Fig. 4b). The absence of Pd atoms outside HEFO lattice (Fig. 4a) and around 6.44% surface Pd atoms determined by CO chemisorption (0.0644 μmol CO/μmol Pd, Supplementary Table 1) together indicate that the Pd atoms have been incorporated into both surface and bulk HEFO phase in Pd$_1$@HEFO, consistent with the reported phenomena that single platinum-group metal atoms prefer to substitute cerium atoms of CeO$_2$ rather than adsorb on its surface[28]. To confirm the electronic structure and coordination state of Pd in Pd$_1$@HEFO, the X-ray absorption near-edge structure (XANES) and EXAFS measurements are performed at the Pd K-edge. XANES spectra show that the Pd K-edge absorption edge for Pd$_1$@HEFO located between that of Pd foil and PdO (Fig. 4d), revealing the valence state of Pd is between 0 and +2, which is lower than +4 of the surface Pd from XPS. This is probably attributed to the fact that the surface Pd atoms are more likely to be contacted with oxygen and be oxidized at high temperatures. Fourier-transformed $k^3$-weighted EXAFS spectra (Fig. 4e and Supplementary Table 3) exhibit the obvious Pd–Pd (bond length = 2.74 Å) and Pd–O–Pd

(bond lengths = 3.06 and 3.44 Å) features for Pd foil and PdO references, respectively, which are both absent in Pd$_1$@HEFO. As an alternative, the bond lengths at 3.01 and 3.26 Å corresponding to Pd–O–Zr and Pd–O–M (M = Ce/La) are identified in Pd$_1$@HEFO (Supplementary Table 3), illustrating the existence of isolated Pd$^{x+}$ (0 < x < 2) in proximity to Zr, Ce, or La atoms[29–31]. The wavelet transform plot (Fig. 4f) of Pd$_1$@HEFO shows the wavelet transform maximum at ~10 Å$^{-1}$, which corresponds to the Pd–O–Zr and Pd–O–M (M = Ce/La) bonding by comparing Pd foil and PdO counterparts and the intensity maxima of Pd–O–Ti at ca. 7 Å$^{-1}$ and Pd–O–Hf at above 12 Å$^{-1}$ [32,33]. Moreover, no intensity maxima corresponding to Pd−Pd and Pd–O–Pd is found, which matches well with the EXAFS fitting results in R space. Consequently, a possible schematic model of Pd$_1$@HEFO (220) is shown in Fig. 4c.

Operando diffuse reflectance infrared Fourier transform spectroscopy (DRIFTS) upon CO adsorption in Fig. 5a is then employed to examine the coordination environment of Pd in Pd$_1$@HEFO and Pd@CeO$_2$. To strengthen the CO adsorption on Pd sites, HEFO, Pd$_1$@HEFO, and Pd@CeO$_2$ are reduced in situ at 250 °C in the DRIFTS cell before exposure to CO flow. For HEFO, the absence of FTIR bands in Fig. 4a suggests the surface of HEFO is not capable of adsorbing CO molecules. In comparison, CO frequencies between 2200 and 2000 cm$^{-1}$ are clearly seen in Pd$_1$@HEFO, which is assignable to CO molecules linearly adsorbed on single-atom Pd species[34,35]. The bridge and hollow-CO bands are not seen in Pd$_1$@HEFO, indicating that Pd are atomically dispersed on the HEFO host, which agrees with HAADF-STEM and EXAFS results[34,36]. However, the bridge and hollow-CO peaks are obviously observed for Pd@CeO$_2$ because of the aggregation of Pd species, thus further validating the importance of hosts for the formation of isolated Pd species[37]. The obvious agglomeration of Pd in Pd@CeO$_2$ can be also evidenced by EDS-mapping results (Supplementary Fig. 7) and existence of the metallic Pd phase from PXRD pattern (Supplementary Fig. 8), in agreement with the bridge and hollow-CO peaks of DRIFTS. The chemical states of Pd in

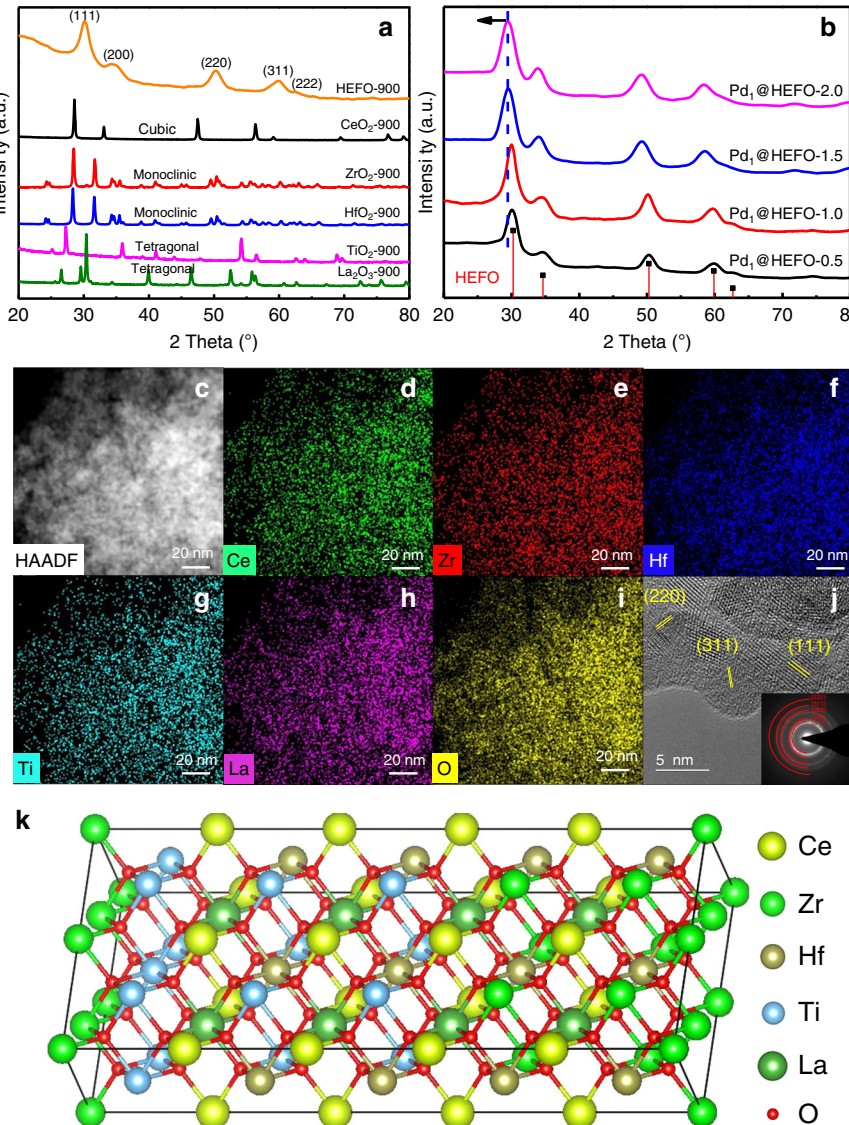

**Fig. 2 Structural characterizations of HEFO and Pd₁@HEFO catalyst. a** PXRD patterns of HEFO, CeO₂, ZrO₂, HfO₂, TiO₂, and La₂O₃ pyrolyzed at 900 °C. **b** PXRD patterns of Pd₁@HEFO-x samples (x is the Pd weight loading). **c** EDS mapping of **d** Ce, **e** Zr, **f** Hf, **g** Ti, **h** La, and **i** O for HEFO. **j** HRTEM image and corresponding FFT pattern (inset) of HEFO. **k** Schematic model of HEFO. Source data are provided as a Source data file.

Pd@CeO₂ and Pd₁@HEFO are both dominated by Pd⁴⁺ based on XPS results (Supplementary Fig. 6), which suggests their similar Pd–O coordination number though totally different Pd micro-environment. As is well known, the phase stabilization is a process determined by combination of the enthalpy (H) and entropy (S) effects, which are temperature- and composition-dependent. Compared with the Pd@CeO₂, Pd₁@HEFO with enhanced compositional complexity provides a higher molar configurational entropy, especially for equimolar cations, which then potentially decreases the Gibbs free energy according to the equation $(\Delta G = \Delta H - T\Delta S)$. This means that the formation of $(Pd_yCeZrHfTiLa)O_x$ solid solution is an entropy-dominated process, whereas the decreased configuration entropy induces the dissociation of $(Pd_yCe)O_x$ as an enthalpy-driven process. To prove this hypothesis, the PXRD patterns of Pd@ZrO₂, Pd@La₂O₃, Pd@HfO₂, Pd@TiO₂, ternary Pd@CeZrTiO$_x$, and quaternary Pd@CeZrHfTiO$_x$ samples synthesized by the same method are also collected (Supplementary Fig. 8). The diffraction peaks ascribed to metallic Pd and/or PdO can be observed, further confirming the importance of the high configurational entropy on stabilizing the Pd single atoms. We also synthesized

the Pd/HEFO-p (single Pd atoms dispersed on HEFO carrier) by a post-deposition method, where the single-atom structure can be confirmed by XRD pattern and CO-DRIFTS spectra in Supplementary Fig. 9. Unfortunately, the sintering and aggregation of Pd on HEFO can be obviously observed in Pd/HEFO-p-900 after post-treatment at 900 °C, which might be ascribed to the excessive Pd density on HEFO surface.

**Catalytic performance.** The oxidation of CO, a key reaction in automotive emission abatement, has been extensively investigated in the past decades[38–41]. Therefore, the light-off curves of CO oxidation are measured to evaluate the catalytic efficiency of our as-obtained samples. As depicted in Fig. 5d, HEFO exhibits an inferior catalytic activity of CO oxidation with a high onset temperature of 230 °C. After doping 1 wt% Pd, Pd₁@HEFO shows a dramatically enhanced reactivity with the onset temperature as low as ~80 °C and complete CO oxidation at 170 °C. In comparison, the onset temperature and $T_{100}$ over Pd@CeO₂ are 223 and 253 °C, respectively, much higher than those of Pd₁@HEFO. It is generally accepted that metal on reducible carrier (CeO₂)

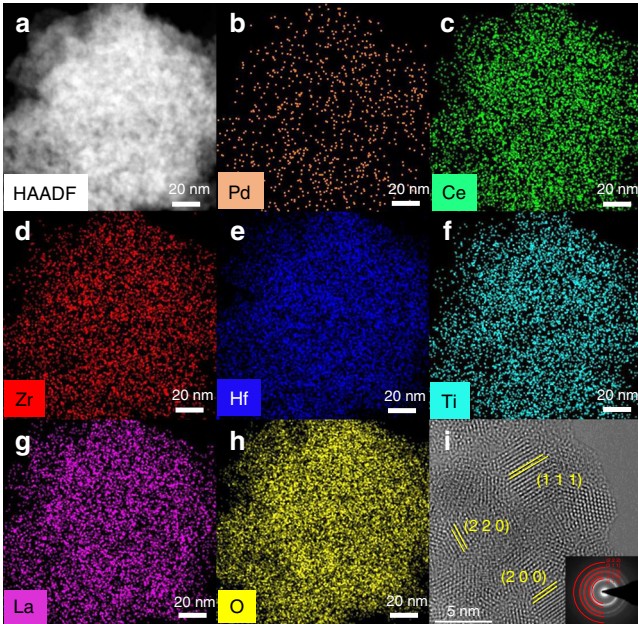

**Fig. 3 Microscopic characterizations of Pd₁@HEFO. a** EDS-mapping image of **b** Pd, **c** Ce, **d** Zr, **e** Hf, **f** Ti, **g** La, and **h** O for Pd₁@HEFO; **i** HRTEM image and corresponding FFT pattern (inset) of Pd₁@HEFO.

follows a Mars–van Krevelen reaction mechanism, where CO adsorbed on metals reacts with active lattice O from $CeO_2$ to form oxygen vacancies[42,43]. Therefore, the reducibility of the surface lattice O from the carrier plays an important role in catalytic CO oxidation. For Pd@CeO₂, the only reduction peak observed at around 250 °C is assigned to the reduction of Pd–O–Pd ($PdO_x$) and the surface lattice oxygen in $CeO_2$[44,45]. In contrast, the surface lattice O of HEFO in the vicinity of Pd (Pd–O–M) is easier to be activated for Pd₁@HEFO compared with Pd–O–Pd for Pd@CeO₂, which offers a reduction temperature centered at 160 °C. The enhanced reducibility of the surface lattice O in Pd₁@HEFO should be the main cause for its higher catalytic activity. Moreover, XPS spectra of O 1 s in Fig. 5c show that the ratio of the surface chemisorbed oxygen ($O_\beta$, 530.4 eV) to the support's lattice oxygen ($O_\alpha$, 528.8 eV) of Pd₁@HEFO is about twofold higher than that of Pd@CeO₂, which might be ascribed to the existence of more under-coordinated metal cations due to the incorporation of Pd into HEFO[46,47], consistent with the surface concentration of $Ce^{3+}$ in Supplementary Fig. 10. These as-formed under-coordinated sites, namely oxygen vacancies, in Pd₁@HEFO would render more habitation for $O_2$ dissociation adsorption[41]. Combined CO-TPR with O 1s XPS results, the improved reducibility of partial surface lattice oxygen in vicinity of Pd and enhanced oxygen vacancies in Pd₁@HEFO should be the main cause for its superior catalytic performance. The apparent activation energies ($E_a$) of Pd@CeO₂ and Pd₁@HEFO were calculated, as shown in Fig. 5e. Pd₁@HEFO has an $E_a$ value of 43.40 kJ/mol, which is much lower than that of Pd@CeO₂ (72.21 kJ/mol), further identify the advantage of Pd₁@HEFO catalyst. As shown in Fig. 5f, the cycled measurement of CO oxidation over Pd₁@HEFO and time-on-stream test at 170 °C (inset) reveal the outstanding stability of Pd₁@HEFO. Since water vapor is usually present in vehicle exhaust, we thereafter examine the hydrothermal stability of both Pd@CeO₂ and Pd₁@HEFO catalysts by treating them at 750 °C for 10 h before the activity test. After hydrothermal treatment, the textural and structural properties of Pd₁@HEFO–HA remain almost unchanged (Supplementary Table 2 and Fig. 5). Correspondingly, all elements are

still uniformly dispersed without agglomeration and sintering of Pd species from the EDS-mapping result in Supplementary Fig. 11. More importantly, the absence of the bridge and hollow-CO peaks further validates that single-atom Pd in Pd₁@HEFO–HA is stable under hydrothermal conditions (Supplementary Fig. 12). Consequently, the complete conversion temperature of CO barely changes after the hydrothermal treatment in Fig. 5d and even 10 vol% $H_2O$ in the feed gas (Pd₁@HEFO-H₂O, Supplementary Fig. 13). The increase of low-temperature catalytic activity of Pd₁@HEFO–HA and Pd₁@HEFO-H₂O (Supplementary Fig. 13) is probably ascribed to the formation of activated surface chemisorbed oxygen ($O_\beta$) on HEFO, reported in the previous study[41]. However, the catalytic activity of Pd@CeO₂–HA obviously decreases compared with its fresh counterpart due to the existence sintering of Pd species, evidenced by the decreased linear-CO and increased bridge and hollow-CO peaks of Pd@CeO₂–HA (Supplementary Fig. 12). In addition, Pd₁@HEFO not only exhibits better thermal and hydrothermal stability than Pd@CeO₂ in this work, but also shows better or comparable performance relative to other reported representative catalysts of CO oxidation[20,41,48,49]. More importantly, Pd₁@HEFO exhibits simultaneously outstanding oxidation activities of CO, $C_3H_6$, and NO at a high gas-hourly space velocity (GHSV) of 200,000 mL gcat⁻¹ h⁻¹ (Supplementary Fig. 14a), although the $T_{100}$ of CO oxidation shifts to ca. 260 °C due to the co-presence of $C_3H_6$ and NO at a high GHSV. The catalytic performance of Pd₁@HEFO is comparable to Pt/CeO₂–SiAlOₓ regarded as a candidate of diesel oxidation catalyst (DOC)[50] and Pt/CeO₂[40]. Moreover, no obvious deactivation of CO, $C_3H_6$, and NO oxidation can be observed over Pd₁@HEFO even after 10 h of reaction in Supplementary Fig. 14b, implying that Pd₁@HEFO shows a good DOC activity and stability. PXRD patterns of Pd₁@HEFO treated in $H_2$ at different temperatures (Supplementary Fig. 15) suggest that Pd atoms in Pd₁@HEFO are not stable in reductive atmosphere. As a result, our Pd₁@HEFO is more suitable for the oxidation reaction under the oxygen-rich conditions, such as catalytic destruction of pollutions emitted from diesel engines. These illustrate that our Pd₁@HEFO possesses not only outstanding low-temperature CO oxidation activity but also excellent resistance to hydrothermal degradation as a candidate of DOC, thus possibly tolerating the harsh conditions during exhaust treatment of diesel engines.

## Discussion

In summary, we have developed a solid-state strategy to synthesize a sintering-resistant Pd single-atom catalyst stabilized on HEFO (Pd₁@HEFO). The as-synthesized single-atom Pd catalyst displays not only superior CO oxidation activity but also outstanding resistance to thermal and hydrothermal degradation compared with Pd@CeO₂ counterpart prepared using the same method. The choice of host in this work plays a paramount role on the synthesis of single-atom Pd catalysts, which can only be realized with HEFO as the carrier because of its maximum configurational entropy. This trait induces Pd to be incorporated into the HEFO sublattice during the mechanochemical-assisted preparation process, and the above process cannot be accomplished with CeO₂ as an alternative carrier. This work provides a solvent-free entropy-driven methodology for the synthesis of SACs and may guide the development of next-generation SACs.

## Methods

**Synthesis of HEFO and Pd₁@HEFO.** Ce(OOCCH₃)₃ (AR, ACROS, USA), ZrCl₄ (98%⁺, Alfa Aesar, USA), HfCl₄ (97.0%, ACROS, USA), La(NO₃)₃ (AR, ACROS, USA), TiOSO₄ (99.0%, ACROS, USA), K₂PdCl₆.xH₂O (99.9%, ALDRICH, USA), and fumed silica (99.9%, ALDRICH, USA) equaled to the total weights of the above five metal precursors were massed and added to a commercially available 25 mL

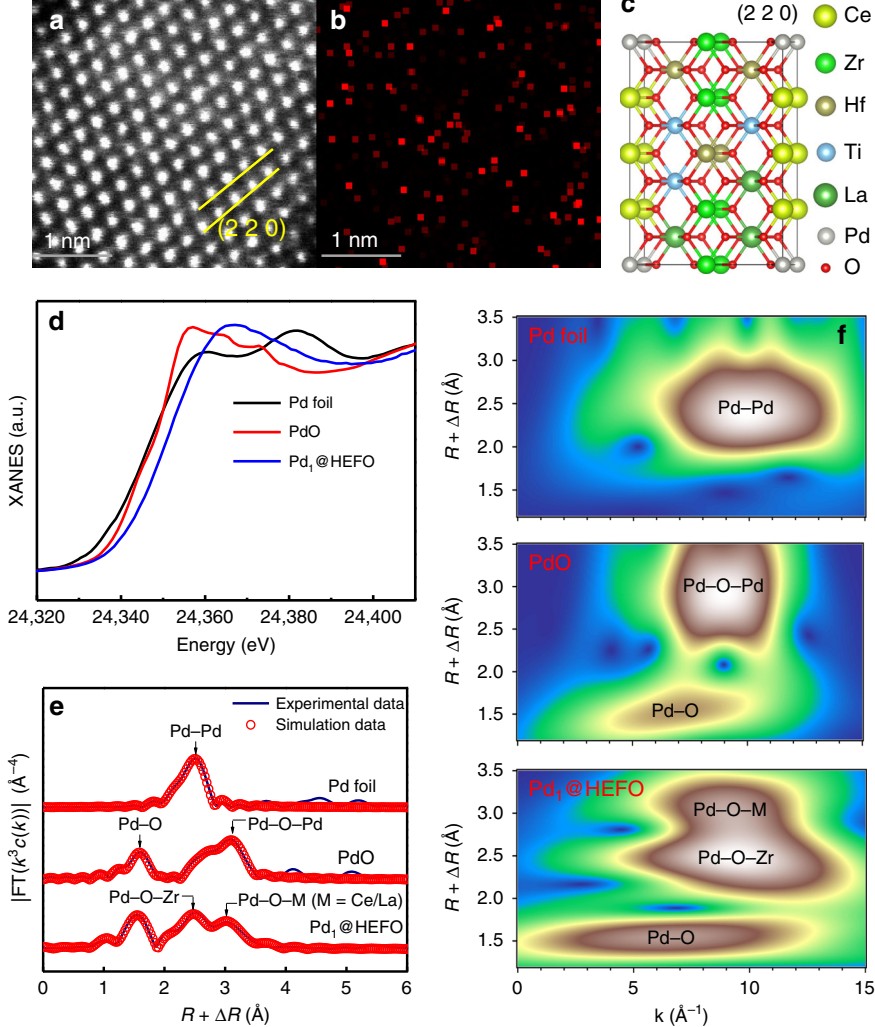

**Fig. 4 Electronic properties of Pd₁@HEFO. a** HAADF-STEM image of Pd₁@HEFO and **b** corresponding EDS mapping of Pd. **c** Schematic model of Pd₁@HEFO (220). **d** XANES spectra at the Pd K-edge. **e** The $k^3$-weighted Fourier transforms of Pd K-edge EXAFS spectra, and **f** the wavelet transforms from experimental data for Pd₁@HEFO, PdO, and Pd foil. Source data are provided as a Source data file.

screw-capped zirconia vial reactor along with five zirconia balls. The reactor was placed in a high-speed vibrating ball miller (1800 rounds min⁻¹, 300 W motor power) and the mixtures were ball milled twice for 30 min each time. The resulting powder product (Pd@CeZrHfTiLaO$_x$@SiO$_2$) was pyrolyzed at 900 °C with a heating rate of 2 °C min⁻¹ for 4 h under air in a tube furnace and cooled down the room temperature, to obtain the sample named as Pd₁@HEFO@SiO$_2$, and then it was stirred in 2.5 M NaOH for 8 h at room temperature and then washed with deionized water, repeating the process four times to remove the silica template, followed by drying at vacuum conditions at 40 °C overnight, finally achieved the porous Pd@HEFO samples. Pd₁@HEFO with different Pd loading (0.5, 1.0, 1.5, and 2.0 wt% Pd) were prepared according to the same above steps, denoted as Pd₁@HEFO-0.5, Pd₁@HEFO-1.0, Pd₁@HEFO-1.5, Pd₁@HEFO-2.0, respectively. Pd₁@HEFO-1.0 was chosen to be as the representative sample to investigate the role of HEFO on the formation of the stabled Pd SACs, which was further briefly labeled as Pd₁@HEFO. HEFO sample was prepared by the same steps as Pd₁@HEFO, without K$_2$PdCl$_6$. Pd@CeO$_2$, Pd@ZrO$_2$, Pd@HfO$_2$, Pd@TiO$_2$, Pd@La$_2$O$_3$, Pd@CeZrTiO$_x$, and Pd@CeZrHfTiO$_x$ were also prepared by the same steps as Pd₁@HEFO. Pd₁@HEFO and Pd@CeO$_2$ samples were treated under 10 vol% water vapor in N$_2$ at 750 °C for 10 h and named as Pd₁@HEFO–HA and Pd@CeO$_2$–HA. Pd₁@HEFO sample was also treated under 2 vol% H$_2$ in He at 250 °C for 2 h and named as Pd₁@HEFO-H$_2$.

**Synthesis of Pd/HEFO-p**. The post-deposition of Pd atoms onto HEFO was obtained by the following typical steps: 0.5 g the pre-synthesized HEFO powder was dispersed in deionized water with rigorous stirring. An appropriate amount of H$_2$PdCl$_6$ (K$_2$PdCl$_6$ dissolved in diluted hydrochloric acid, corresponding to a Pd loading of 0.5 wt%) solution was added dropwise into the HEFO solution under magnetically stirring[51]. After continuing stirring for 5 h and followed by aging for 5 h, the suspension was filtered and washed with deionized water for several times,

and dried at 60 °C under vacuum[52] and then calcined at 300 and 900 °C in air for 2 h, respectively, denoted as Pd/HEFO-p and Pd/HEFO-p-900.

**Characterizations**. X-ray diffraction (XRD) data were recorded on a PANalytical Empyrean diffractometer with a Cu-Kα radiation source in the $2\theta$ range of 20–80°. The nitrogen adsorption and desorption isotherms were measured at 77 K under a Gemini VII surface-area analyzer. Samples were degassed for 6 h under N$_2$ at 160 °C prior to the measurement. High-resolution transmission electron microscopy (HRTEM), high-angle annular dark-field scanning transmission electron microscopy (HAADF-STEM) and the corresponding energy-dispersive X-ray spectroscopy (EDS) were conducted on an aberration-corrected Titan S 80-300 (FEI) with an accelerating 300 kV voltage.

The X-ray absorption spectra (XAS) were collected on the beamline BL01C1 in National Synchrotron Radiation Research Center (NSRRC), with electron energy of 1.5 GeV and a beam current between 100 and 200 mA, and XAS data were collected at treatment temperatures. The radiation was monochromatized by a Si (111) double-crystal monochromator. XANES and EXAFS data reduction and analysis were processed by Athena software. More details on the fitting process are given in the figure legend or table footnote.

In situ diffuse reflectance infrared Fourier transform spectra (DRIFTS) were collected at 30 °C with a Thermo Nicolet 6700 spectrometer equipped with a mercury cadmium telluride (MCT) detector accumulated at a resolution of 4 cm⁻¹ in 100 scans. Fifty milligrams of samples were preconditioned in situ at 400 °C under He (99.999%) for 30 min and then cooled to 250 °C to be reduced for 30 min in 5 vol% H$_2$/N$_2$ (30 mL min⁻¹). Finally, the sample was cooled to 30 °C prior to recording the background spectra and exposed to 1 vol% CO/N$_2$ flow until no change of the spectrum, and the CO adsorption spectrum was collected after purging in He.

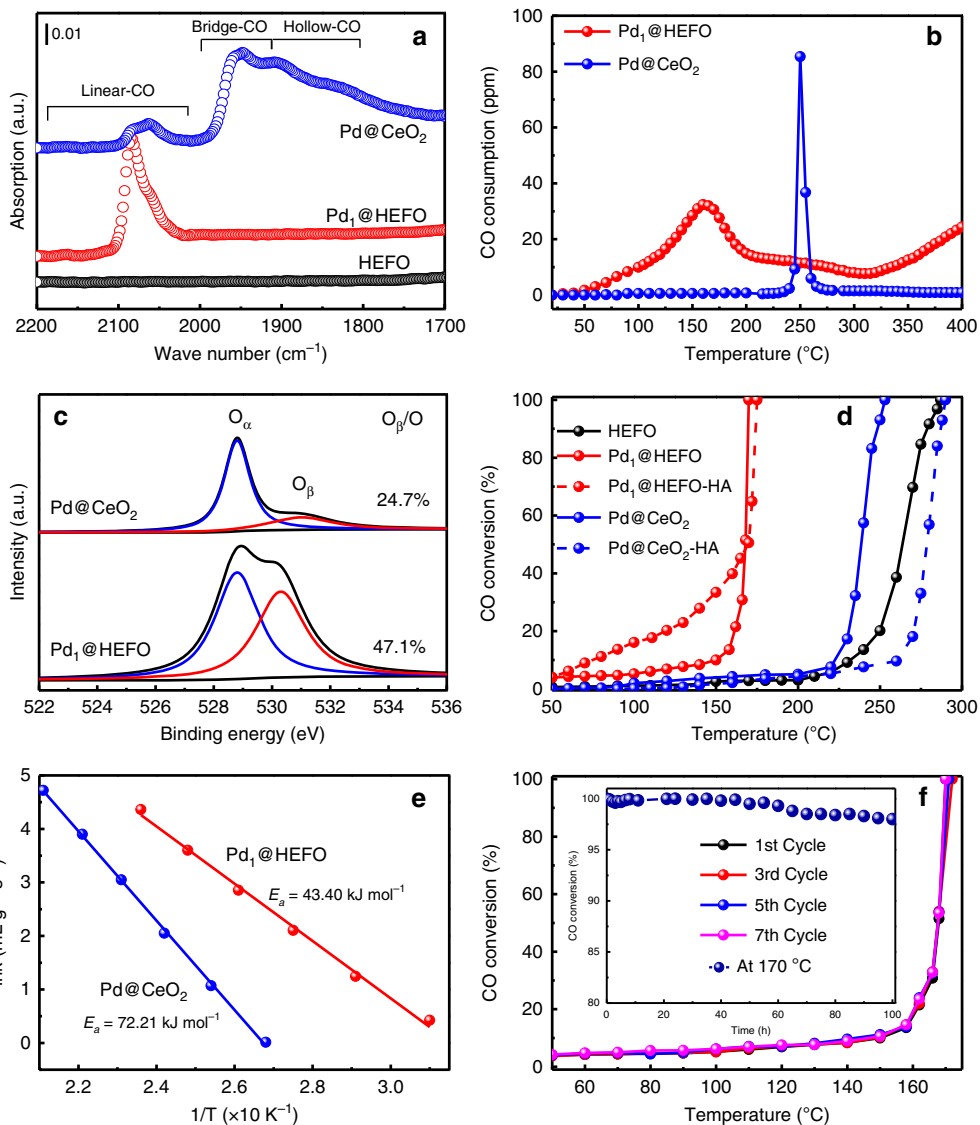

**Fig. 5 Catalytic performances of CO oxidation for Pd₁@HEFO and Pd@CeO₂. a** CO-DRIFTS results of HEFO, Pd₁@HEFO, and Pd@CeO₂. **b** CO-TPR of Pd₁@HEFO and Pd@CeO₂. **c** XPS profiles of O 1s for Pd₁@HEFO and Pd@CeO₂. **d** CO oxidation of catalytic performance of different catalysts before (solid lines) and after (dashed lines) hydrothermal treatments. Reaction conditions: A catalyst loading of 20 mg, and 1 vol% CO balance in air at a gas-hourly space velocity of 40,000 mL gcat⁻¹ h⁻¹. **e** Arrhenius plots of Pd₁@HEFO and Pd@CeO₂. **f** the cycled measurement of CO oxidation over Pd₁@HEFO and its stability at 170 °C (inset). Source data are provided as a Source data file.

X-ray photoelectron spectroscopy (XPS) measurements were performed with a Kratos XSAM-800 Kα instrument. Peak fitting was performed using CasaXPS software (v 2.3.18). The C 1s peak (284.6 eV) was used for the calibration.

The pulse CO chemisorption was measured by Micromeritics AutoChem II 2920. Hundred milligrams of sample was reduced in H₂ at 400 °C for 1 h and then the sample was cleaned by He for 1 h with a flow rate of 30 ml min⁻¹. Then, the CO was injected (11.10 μmol/pluse) into the catalyst in a flow of He until the amount of CO uptake was saturated. The exposed Pd atoms were evaluated from the amount of CO consumption assuming 1 molecule of CO adsorbed per surface Pd atom[32].

CO temperature-programmed reduction (CO-TPR) experiments were conducted by the FTIR spectrometer (Antaris IGS, Nicolet). The powder catalyst put in flow reactor device. Fifty milligrams of sample was put in a fixed bed quartz and pretreated at 450 °C (10 °C min⁻¹) in He for 1 h and then cooled down to room temperature. Then the sample was heated to 500 °C (5 °C min⁻¹) under an atmosphere of 1 vol% CO in He (200 mL min⁻¹).

O₂-temperature-programmed desorption (O₂-TPD) experiments were performed on a PX200 apparatus equipped with a TCD. Fifty milligrams of the sample was pretreated in a flow of N₂ (20 mL min⁻¹) at 400 °C for 30 min and then cooled to 80 °C. The adsorption was carried out by 5 vol% O₂–95 vol% N₂ (30 mL min⁻¹) at 80 °C for 30 min. Then the samples were swept at 80 °C for

60 min by He. The desorption was detected from 80 to 400 °C with a heating rate of 5 °C min⁻¹.

**CO oxidation activity measurement.** CO oxidation reaction was performed in the same manner as our previous work[5]. The outlet concentrations of CO and CO₂ were analyzed using an on-line gas chromatograph (Buck Scientific 910) equipped with a dual molecular sieve/porous polymer column (Alltech CTR1) with a thermal conductivity detector.

The apparent activation energy was calculated using the Arrhenius law ($k = A\exp(-E_a/RT)$). $E_a$ was obtained from the slope of the linear plot of $\ln(k)$ versus $1000/T$ and the value of $k$ was calculated using the following equation:

$$k = (V \times X)/m$$

Where, $V$ is the total gas flow (mL·s⁻¹) at temperature $T$ (K), $X$ is the CO conversion (%), and $m$ is the catalyst mass in the reactor (g). In this work, the weights of Pd₁@HEFO and Pd@CeO₂ are both 10 mg.

**DOC catalytic activity measurement.** The DOC catalytic activity evaluation of Pd₁@HEFO was performed according to the previous work[49]. A catalyst loading of 150 mg at a gas-hourly space velocity (GHSV) of 200,000 mL gcat⁻¹ h⁻¹ with the

total gas flow rate of 500 ml min$^{-1}$ was used. The outlet flows of CO, $C_3H_6$, NO, and $NO_2$ were analyzed using a Nicolet Antaris IGS-6700 gas analyzer (Thermo Fisher Scientific, USA).

## Data availability

All the data that support the plots within this paper and its Supplementary Information are available from the corresponding author upon reasonable request. Source data are provided with this paper.

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

## Acknowledgements

Z.Z., J.L., C.D., H.C., and S.D. were supported by the U.S. Department of Energy, Office of Science, Office of Basic Energy Sciences, Chemical Sciences, Geosciences, and Biosciences

Division, Catalysis Science program. H.X., S.X., Q.L., Y.J., and Y.W. were supported by the National Natural Science Foundation of China (Nos. 21802099, 21972098), National Key Research & Development Program of China (No. 2016YFC0204901), National Engineering Laboratory for Mobile Source Emission Control Technology (No. NELMS2017A06) and Sichuan Science and Technology Program (No. 2018GZ0401). Y.C. and J.W. were supported by the National Natural Science Foundation of China (No. 21673146). H.X. also thanks the China Scholarship Council for financial support as a visiting scholar. Authors thank Dr. Miaomiao Liu in East China University of Science and Technology for giving the comments on the Scheme and Ms. Jia Li in Sichuan University for the models of HEFO and $Pd_1@HEFO$. Ceshigo Research Service for agency STEM and XAS, www.ceshigo.com.

## Author contributions

S.D. conceived the research idea and H.X., Y.W., and Y.C. designed the experiments. H.X. performed all the experiments and analyzed all the data. Z.Z. analyzed the data of TEM, XAS, and CO-DRIFTS. J.L. and H.C. carried out XRD and XPS tests. C.D. took part in the synthesis of samples. S.X. and Q.L. carried out the CO-TPR and $O_2$-TPD tests. Y.J. and J.W. discussed the results and commented on the paper. H.X., Z.Z., C.D., Y.W., Y.C., and S.D. co-wrote and revised the paper.

## Competing interests

The authors declare no competing interests.
