## [Peer Review File · Nature Communications]

Reviewers' comments:

Reviewer #1 (Remarks to the Author):

In this work Xu and co-authors reported the synthesis and characterization of multicomponent oxide, the so-called high-entropy oxide, supported highly stable Pd single-atom catalyst (SAC) as well as its catalytic performance in CO oxidation. The SAC was synthesized by a ball milling method and characterized by a combination of aberration corrected STEM, EXAFS as well as DRIFT. Although the former two characterization results are less convincing due to the low contrast between Pd and support elements, the DRIFT spectrum is very impressive and convincing. Therefore, I have no doubt about the formation of highly stable isolated Pd single atoms on the high-entropy oxide support. The high activity and stability of the catalyst in CO oxidation is also attractive. However the experimental data as well as scientific insights into the catalyst preparation and stability are lacking and remain concern. In whole I think the results are interesting. I recommend it for publication after addressing the following questions and concerns.

1. What does the term “intrinsically stable” mean? Do the authors mean that single atoms are more stable than their nanoparticle/nanocluster counterpart or mean single atoms are stable at elevated temperatures? I think authors should clearly illustrate this. Actually the former had not been demonstrated as no comparison with NPs/NCs on identical support was performed in this work while the latter is only demonstrated in oxidative atmosphere.
2. The Pd SACs were indeed successfully fabricated on a high-entropy oxide support which shows good stability. However, whether the good stability is due to the high configurational entropy remains unaddressed. For example, He et al have recently reported (Cell Reports Physical Science 2020, 1, 100004) that single oxide supported SAC can be synthesized by the same method which suggests that the high-entropy oxide is not indispensable. Therefore, in this work the formation of stable Pd SACs might originate from the strong interaction between Pd and one of the five elements rather than the high configurational entropy. This may be just like the strong interaction between Pt-CeO₂ (Science 2016, 353, 150) and Pt-Fe₂O₃ (Nat Commun, 2019, 10, 234)
3. In SACs prepared by this method I believe some of the Pd atoms located in the bulk of the support instead of on the surface. So the surface exposed Pd atoms should be quantified by chemical adsorption. In addition, how about the post-deposition of Pd atoms onto the pre-synthesized high-entropy oxide support? Actually it would be more attractive if the post-deposited Pd single atoms are still stable.
4. How about the stability of this catalyst (or the post-deposited catalyst) in reductive atmosphere? What temperature can they stand for in reductive atmosphere?
5. The high-entropy oxide dominated by four 4+ cation and one 3+ cation (Zr³⁺). The latter may result in the formation of vacancies/defects. Is it possible that the defects play a critical role in stabilizing the Pd single atoms?
6. XANES result suggested Pd existed as 4+ while XPS characterization evidenced Pd existed in 2+ or lower. Authors should explain this discrepancy.
7. EXAFS fitting results suggested that Pd bonding to Zr and Ce/La through O, why no Pd-O-Ti or Pd-O-Hf bonding?
8. In Fig 1b, Pd₁@HEFO should be Pd₁@HEFO-1.0

9. In the caption of Fig4, 4h should be 4f

Reviewer #2 (Remarks to the Author):

This is a generally interesting work, describing the synthesis of a single atom Pd catalyst on a high entropy fluorite, in particular mixed Ce, Zr, Hf, Ti, La oxide, doped with Pd ions. The material was characterized thoroughly and evaluated as a catalyst for the CO oxidation, which is of a high interest for the treatment of vehicle exhaust gas.

The work highlights the use of the mixed fluorite as a novel matrix to stabilize the Pd ions in the lattice, avoiding sintering and formation of the less catalytically active clusters of PdOx species. It further highlights the activity of the catalyst for its ability to perform CO oxidation at low temperatures (50 % conversion at ca. 130 oC), in comparison to a similarly Pd-doped system based on CeO₂ only. Nevertheless, the importance of the claimed novelties is significantly restricted by the fact that sintering-resistant single atom catalysts for various catalytic reactions have already been reported in the literature. Furthermore, single atom catalysts active for CO oxidation at very low temperatures have been also reported in several articles. The Pd loading of the present work of 0.25 at. % helps the stabilization of Pd in the fluorite lattice, but it makes difficult to highlight its superior stabilization properties with respect to previous works.

In particular, in J. Catal. 1995, 153, 304, W. Liu and M. Flytzani-Stefanopoulos reported the mixed Ce-La fluorite doped in the lattice with Cu, which presented 50 % conversion even at 80 oC, for the same reaction. Au doping was also reported, which showed even room temperature activity. Although at that time the exact configuration was not clear, later studies showed that 0.2-0.9 at % contents of Au exist solely as dopant ions in the lattice of the La-containing ceria (Fu et al. Appl Catal B-Environ. 2005, 56, 57). The substitution of Ce⁺⁴ with the lower valence La⁺³, was also recognized for its importance in creating sites for lattice doping with metals ions.

Stable Pd incorporation in the lattice, as in Ce_{0.93}Pd_{0.02}Cu_{0.05}O_{2-δ} (Catal. Sci. Technol. 2016, 6, 8104) has been also reported with excellent performance for CO oxidation. Further single metal atom catalysts for CO oxidation have been reported with very high activities (Rh in ceria, Chem. Mater. 2004, 16, 11, 2317) (Pd and Pt in metal oxides, ACS catalysis 2019, 9, 1595) (Pd in La⁺³ containing alumina Nat Commun 2014, 5, 4885) (Pt, Ru and Co single atoms, on carbon, C₃N₄ and TiO₂ substrates Nat. Nanotechnol. 2019, 14, 851).

Methods for preparing sintering-stable single atom catalysts have been also reported for other types of oxidation reactions (Nat. Commun. 2019, 10, 234; Nat Commun 2020, 11, 335; J. Am. Chem. Soc. 2019, 141, 18, 7283).

Therefore, it is not quite clear what are the new insights/findings which the present work conveys.

Some other points:

In line 144, electron deficient Pd⁺⁴ is found for the catalyst by XPS, and later an intermediate oxidation between 0 and +2 (line 159) is found from XANES spectra. Why is that?

The stability of the catalyst was tested, but low conversion was obtained (30%, Figure 4f inset). How was the stability at higher conversions? The deactivation test by water vapour was performed ex-situ, but for

practical reasons water could be present during the catalytic reaction.

In the current form, the manuscript is not acceptable for publication in Nature Communications.

Response to Reviewers

Reviewers' comments:

Reviewer #1 (Remarks to the Author):

1. What does the term “intrinsically stable” mean? Do the authors mean that single atoms are more stable than their nanoparticle/nanocluster counterpart or mean single atoms are stable at elevated temperatures? I think authors should clearly illustrate this. Actually the former had not been demonstrated as no comparison with NPs/NCs on identical support was performed in this work while the latter is only demonstrated in oxidative atmosphere.

Response: Single Pd atoms on HEFO are at least more thermodynamically stable than their nanoparticles (NPs) counterpart since no Pd NPs are found after long-term treatment at 900 °C. Therefore, we utilize the term “intrinsically stable” for our Pd₁@HEFO catalyst and want to tell it apart from the conventional single atom catalysts, which usually require a very low post-treatment temperature to attain the single atom dispersion. As seen in question 3, we synthesized the single Pd atoms dispersed on HEFO catalyst (Pd/HEFO-p) by a post-deposition method. Unfortunately, the sintering and aggregation of Pd in Pd/HEFO-p can be obviously observed in Pd/HEFO-p-900 after treatment at 900 °C, which means HEFO itself cannot stabilize Pd atoms at high temperatures. Therefore, stable Pd atoms seems to be only attained during the formation of the HEFO phase with the form of Pd-HEFO solid solution, which further explains the uniqueness of our method. The stability test of Pd₁@HEFO catalyst under reductive atmosphere has also been included and discussed (seen question 4).

Characterization results reveal that Pd atoms are incorporated into HEFO sublattice with Pd valence state between 0 and +2, forming a stable Pd-O-M bonds (M=Ce, Zr, La). In comparison, Pd@CeO₂, Pd@ZrO₂, Pd@La₂O₃, Pd@HfO₂, Pd@TiO₂, ternary Pd@CeZrTiO_x and quaternary Pd@CeZrHfTiO_x synthesized by the same method (see question 2) affords the diffraction peaks ascribed to Pd and/or PdO_x clusters due to its relatively lower configurational entropy. Therefore, the “intrinsically stable” means that the high-entropy fluorite oxides ((CeZrHfTiLa)O_x) possess higher configurational entropy than other counterparts. “intrinsically stable” has been revised as “Entropy-Stabilized” in the title to avoid ambiguity. The revised title is as follows:

“Entropy-Stabilized Single-Atom Pd Catalysts via High-Entropy Fluorite Oxide Supports”.
(Page 1 in the revised manuscript)

2. The Pd SACs were indeed successfully fabricated on a high-entropy oxide support which shows good stability. However, whether the good stability is due to the high configurational entropy remains unaddressed. For example, He et al have recently reported (Cell Reports Physical Science 2020, 1, 100004) that single oxide supported SAC can be synthesized by the same method which suggests that the high-entropy oxide is not indispensable. Therefore, in this work the formation of stable Pd SACs might originate from the strong interaction between Pd and one of the five elements rather than the high configurational entropy. This may be just like the strong interaction between Pt-CeO₂ (Science 2016, 353, 150) and Pt-Fe₂O₃ (Nat Commun, 2019, 10, 234).

Response: It is no doubt that Pd-ZnO (Cell Reports Physical Science 2020, 1, 100004), Pt-CeO₂ (Science 2016, 353, 150) and Pt-Fe₂O₃ (Nat Commun, 2019, 10, 234) showed the single metal dispersion under the specific preparation conditions. For the Pd-ZnO sample prepared by the mechanochemical approach, the metal precursors were limited to acetylacetonate salt, and longer ball milling time (10 h) as well as the relatively lower calcination temperature (400 °C) were required to realize the formation of single Pd atoms. Additionally, the atom trapping protocol reported first from Datye and co-workers (Science 2016, 353, 150) is very useful for preparing Pt single atoms at high temperature such as 800 °C because of the evaporation of PtO₂ and reducible CeO₂ or FeO_x that can continuously bind the mobile species. But this method wasn't extended to other metals like Pd due to the differences in chemical properties of different metals to the best of our knowledge. In order to confirm the role of high configurational entropy in this work, we included Pd@ZrO₂, Pd@La₂O₃, Pd@HfO₂, Pd@TiO₂, ternary Pd@CeZrTiO_x and quaternary Pd@CeZrHfTiO_x samples synthesized by the same method as Pd₁@HEFO and Pd@CeO₂. XRD patterns of all samples were displayed in Supplementary Fig. 8. The diffraction peaks at ~40.2° or ~42.0° ascribed to metallic Pd and PdO, respectively, can be clearly observed in single, ternary and quaternary metal oxides supported Pd catalysts. In sharp contrast, no diffraction peaks attributed to Pd species can be observed in Pd@CeZrHfTiLaO_x (Pd₁@HEFO) from Fig. 1, which is the clear evidence that the formation of Pd SACs in Pd₁@HEFO originates from the high configurational entropy rather than the facile interactions between Pd and one of support elements.

The related content has been added in the revised manuscript as follows: “The obvious agglomeration of Pd in Pd@CeO₂ can be also evidenced by EDS-mapping result (Supplementary Fig. 7c) and existence of the metallic Pd phase from PXRD pattern (Supplementary Fig. 8), in agreement with the bridge and hollow-CO peaks of DRIFTS. PXRD patterns of Pd@ZrO₂, Pd@La₂O₃, Pd@HfO₂, Pd@TiO₂, ternary Pd@CeZrTiO_x and quaternary Pd@CeZrHfTiO_x samples synthesized by the same method are collected. The diffraction peaks ascribed to metallic Pd and/or PdO can be observed, further confirming the importance of the high configurational entropy.” (Page 11 in the revised Manuscript)

Supplementary Fig. 8 PXRD of Pd@CeO₂, Pd@ZrO₂, Pd@La₂O₃, Pd@HfO₂, Pd@TiO₂, Pd@CeZrTiO_x and Pd@CeZrHfTiO_x samples.

3. In SACs prepared by this method I believe some of the Pd atoms located in the bulk of the support instead of on the surface. So the surface exposed Pd atoms should be quantified by chemical adsorption.

Response: Thanks for your suggestion. The pulse CO chemisorption experiment was performed on a Micromeritics AutoChem II 2920 to quantify the surface exposed Pd atoms. The result showed that 14.77 $\mu\text{L g}^{-1}$ of CO adsorbed on Pd₁@HEFO sample. Since only the linear CO band can be detected on Pd₁@HEFO based on CO-DRIFTS spectrum (Fig. 4a), we assume that only

one CO molecule bonded on one surface Pd atom. The calculated value was 0.0644 $\mu\text{mol CO}/\mu\text{mol Pd}$ for Pd₁@HEFO, which means the surface exposed Pd atoms only accounted for 6.44% of the total Pd atoms in Pd₁@HEFO. The results are reasonable since Pd atoms have been incorporated into both HEFO surface and bulk lattice, and the Pd atoms in bulk HEFO phase are not used in reality. In a future study, decreasing the HEFO particle size and attaining more surface Pd atoms should be considered for the synthesis of more efficient catalysts. We have included the relevant information in the manuscript to tell readers the surface Pd contents of our Pd₁@HEFO sample:

“The absence of Pd atoms outside HEFO lattice (Fig. 3a) and around 6.44% surface Pd atoms determined by CO chemisorption (Supplementary Table 1) together indicates the Pd atoms have been incorporated into both HEFO surface and bulk phase in Pd₁@HEFO”. (Page 8 in the revised Manuscript)

In addition, how about the post-deposition of Pd atoms onto the pre-synthesized high-entropy oxide support? Actually it would be more attractive if the post-deposited Pd single atoms are still stable.

Response: We tried to synthesize a Pd/HEFO-p catalyst via a facile adsorption method based on the works from Wu et al. (Nat. Comm. 2020, 11, 335) and Zhang et al. (Nat. Comm. 2019, 10, 4500; ACS Catal. 2015, 5, 6249). The detailed synthesis information is shown in the manuscript, and the XRD and CO-DRIFTS results are shown in the supporting information as new Supplementary Fig. 9:

From the XRD patterns and CO-DRIFTS spectra, the Pd/HEFO-p catalyst synthesized by the post-deposition is the atomically dispersed Pd species without the characteristic of Pd clusters or NPs. However, after calcination at 900 °C in air, a sharp diffraction peak at ca. 40.18° was observed in XRD pattern of Pd/HEFO-p-900 from Supplementary Figure 9a, correspondingly, the bridge-CO and hollow-CO bands are obviously observed for Pd/HEFO-p-900 because of the aggregation of Pd species (Supplementary Figure 9b). Combination with XRD and CO-DRIFTS results of Pd/HEFO-p and Pd/HEFO-p-900, single Pd atoms can be attained on HEFO via a post-deposition method. Unfortunately, the sintering and aggregation of Pd on HEFO can be obviously observed after high-temperature treatment at 900 °C. It illustrated that the phase

transformation of single metal oxides into high-entropy fluorite oxides is crucial for the dispersion of single Pd atoms by constructing the stable Pd-O-M bonds in both surface and bulk HEFO lattice, also known as Pd-HEFO solid solution. While these stable Pd-HEFO solid solution structure seems to be difficult to formed in Pd/HEFO-p via a post-deposition method, resulting in the aggregation of single Pd atoms at 900 °C in air. Instead, single Pd atoms in Pd/HEFO-p might locate outside the HEFO lattice and tend to become mobile and then grow larger at high temperatures. However, more research is still needed to unravel these different but interesting behaviors clearly. We also included the characterizations of Pd/HEFO-p and Pd/HEFO-p-900 and relative discussions in this manuscript:

“We also synthesized the Pd/HEFO-p (single Pd atoms dispersed on HEFO carrier) by a post-deposition method, where the single atom structure can be confirmed by XRD pattern and CO-DRIFTS spectra in Supplementary Fig. 9. Unfortunately, the sintering and aggregation of Pd on HEFO can be obviously observed in Pd/HEFO-p-900 after treatment at 900 °C, which means HEFO itself cannot stabilize Pd atoms at high temperatures. Therefore, stable Pd atoms seems to be only attained during the formation of HEFO phase with the form of the Pd-HEFO solid solution, which further underscores the uniqueness of our method.” (Page 11 in the revised manuscript)

Supplementary Fig. 9 (a) XRD patterns and (b) CO-DRIFTS spectra of Pd/HEFO-p and Pd/HEFO-p-900.

4. How about the stability of this catalyst (or the post-deposited catalyst) in reductive atmosphere? What temperature can they stand for in reductive atmosphere?

Response: To study the stability of Pd₁@HEFO in reductive atmosphere, the sample was treated in 5% H₂/N₂ at 250, 500 and 900 °C for 2 h, respectively. The H₂-treated samples were performed by XRD, as shown in Supplementary Figure 15. No peaks assigned to metallic Pd or PdO_x can be observed in the sample treated at 250 °C, however, a diffraction peak at ca. 40.20° attributed to metallic Pd was detected when the reduced temperature increased to 500 °C and the peak intensity further strengthened at 900 °C. It suggested that single Pd atoms can be stabilized on HEFO in the reductive atmosphere at low temperatures (such as 250 °C), but further increasing reductive temperature resulted in the sintering and aggregation of reduced Pd. The above results suggested that the high-temperature stability of Pd atoms in Pd₁@HEFO is very good in oxidative not reductive atmosphere, which can be explained by Pd-O-M bonds formation in oxidative atmosphere. The reductive atmosphere might inhibit the formation of robust Pd-O-M bonds, thus increasing the mobility of Pd atoms and thus causing the formation of Pd particles.

“PXRD patterns of Pd₁@HEFO treated in H₂ at different temperatures (Supplementary Fig. 15) suggest that Pd atoms in Pd₁@HEFO are not stable in reductive atmosphere, as a result, our Pd₁@HEFO is more suitable for the oxidation reaction under the oxygen-rich conditions, such as catalytic destruction of pollutions emitted from diesel engines.” (Page 13-14 in the revised manuscript)

Supplementary Fig. 15 XRD patterns of Pd₁@HEFO after H₂ treatment at different temperatures for 2 h.

5. The high-entropy oxide dominated by four 4+ cation and one 3+ cation (Zr³⁺). The latter may result in the formation of vacancies/defects. Is it possible that the defects play a critical role in stabilizing the Pd single atoms?

Response: We are not very clear on the role of vacancies in stabilizing the Pd single atoms. In this work, Pd single atoms are stabilized by robust Pd-O-M bonds, which can be proved by TEM, DRIFTS and EXAFS results. The existence of vacancies would lead to the formation of under-coordinated metals in HEFO, but it is unclear what effect these under-coordinated metals have on the formation of Pd-O-M bonds. Sorry, we cannot provide you a definite answer at this stage.

6. XANES result suggested Pd existed as 4+ while XPS characterization evidenced Pd existed in 2+ or lower. Authors should explain this discrepancy.

Response: The obtained binding energy (Supplementary Fig. 6) from XPS is the characteristic of electron-deficient Pd⁴⁺ in Pd₁@HEFO. However, XANES spectra show that the Pd K-edge absorption edge for Pd₁@HEFO is located between that of Pd foil and PdO (Fig. 3d), revealing the valence state of Pd is between 0 and +2. The difference between the valence state of Pd from XPS and XANES results come from the fact that XPS technology characterizes the surface elemental valence state while XANES as a more advanced technology can characterize the bulk elemental states and achieve the average valence of Pd in the bulk. Pd is easier to be oxidized to higher valence on the surface rather than on the bulk, so the valence of surface Pd (XPS) is higher than that of bulk Pd (XANES). Therefore, the difference between the valences of Pd from XPS and XANES is reasonable and not contradictory. The related expressions have been revised as follows:

“XANES spectra show that the Pd K-edge absorption edge for Pd₁@HEFO is located between that of Pd foil and PdO (Fig. 3d), revealing the valence state of Pd is between 0 and +2, which is lower than +4 of the surface Pd from XPS. This is probably attributed to that the surface Pd atoms is more likely to be contacted with oxygen and be oxidized at high temperature.” (Page 9 in the revised manuscript)

7. EXAFS fitting results suggested that Pd bonding to Zr and Ce/La through O, why no Pd-O-Ti or Pd-O-Hf bonding?

Response: The wavelet transform (WT) analysis of the Pd₁@HEFO K-edge spectroscopy was provided in Fig. 3f. It could be observed that the intensity maxima at ca. 10 Å⁻¹ were associated with the Pd-O-Zr and Pd-O-M (M=Ce/La) contributions, respectively, which agrees well with the EXAFS fitting results in R space. On the one hand, the intensity maxima of Pd-O-Ti is located at ca. 7 Å⁻¹ (J. Catal. 2009, 266, 15) due to Ti as a light element while that of Pd-O-Hf is centered at above 12 Å⁻¹ (Phy. Rev. B 1998, 58, 7565) due to Hf as a heavy element, which are inconsistent with the WT plot of Pd₁@HEFO in Fig. 3f. Compared to the PdO reference, the intensity maxima of Pd-O-M for Pd₁@HEFO is closer to that of Pd-O-Pd for PdO, so the atomic number of M is also close to that of Pd. The atomic numbers of six elements in Pd₁@HEFO are increased as follows: Ti (22) < Zr (40) < Pd (46) < La (57) < Ce (58) < Hf (72), so Zr, Ce and La are the atoms that most likely bond to Pd in Pd₁@HEFO. On the other hand, we also checked the coordination environment of Pd-O-Ti, and Pd-O-Hf, but the fitting result does not match well with the experimental data, so the coordination possibilities of Pd-O-Ti and/or Pd-O-Hf in Pd₁@HEFO should be lower. Therefore, we assigned the Pd-O-M into Pd-O-Zr/Ce/La instead of Pd-O-Ti and Pd-O-Hf in this manuscript. The revised expression in the manuscript is:

“The wavelet transform plot (Fig. 3f) of Pd₁@HEFO shows the wavelet transform maximum at approximately 10 Å⁻¹, which corresponds to the Pd–O–Zr and Pd–O–M (M = Ce/La) bonding by comparing Pd foil and PdO counterparts and the intensity maxima of Pd-O-Ti at ca. 7 Å⁻¹ and Pd-O-Hf at above 12 Å⁻¹.³²⁻³³” (Page 9 in the revised Manuscript)

8. In Fig 1b, Pd₁@HEFO should be Pd₁@HEFO-1.0.

Response: Thanks for your suggestion. Pd₁@HEFO has been revised to Pd₁@HEFO-1.0 in Fig. 1b and Supplementary Table 2. And the related expressions in “Method” have also been revised.

9. In the caption of Fig4, 4h should be 4f.

Response: It has been corrected.

Reviewer #2 (Remarks to the Author):

The work highlights the use of the mixed fluorite as a novel matrix to stabilize the Pd ions in the lattice, avoiding sintering and formation of the less catalytically active clusters of PdO_x species.

It further highlights the activity of the catalyst for its ability to perform CO oxidation at low temperatures (50 % conversion at ca. 130 °C), in comparison to a similarly Pd-doped system based on CeO₂ only.

Nevertheless, the importance of the claimed novelties is significantly restricted by the fact that sintering-resistant single atom catalysts for various catalytic reactions have already been reported in the literature. Furthermore, single atom catalysts active for CO oxidation at very low temperatures have been also reported in several articles. The Pd loading of the present work of 0.25 at. % helps the stabilization of Pd in the fluorite lattice, but it makes difficult to highlight its superior stabilization properties with respect to previous works.

In particular, in J. Catal. 1995, 153, 304, W. Liu and M. Flytzani-Stefanopoulos reported the mixed Ce-La fluorite doped in the lattice with Cu, which presented 50 % conversion even at 80 °C, for the same reaction. Au doping was also reported, which showed even room temperature activity. Although at that time the exact configuration was not clear, later studies showed that 0.2-0.9 at % contents of Au exist solely as dopant ions in the lattice of the La-containing ceria (Fu et al. Appl Catal B-Environ. 2005, 56, 57). The substitution of Ce⁺⁴ with the lower valence La⁺³, was also recognized for its importance in creating sites for lattice doping with metals ions. Stable Pd incorporation in the lattice, as in Ce_{0.93}Pd_{0.02}Cu_{0.05}O_{2-δ} (Catal. Sci. Technol. 2016, 6, 8104) has been also reported with excellent performance for CO oxidation. Further single metal atom catalysts for CO oxidation have been reported with very high activities (Rh in ceria, Chem. Mater. 2004, 16, 11, 2317) (Pd and Pt in metal oxides, ACS catalysis 2019, 9, 1595) (Pd in La⁺³ containing alumina Nat Commun 2014, 5, 4885) (Pt, Ru and Co single atoms, on carbon, C₃N₄ and TiO₂ substrates Nat. Nanotechnol. 2019, 14, 851).

Methods for preparing sintering-stable single atom catalysts have been also reported for other types of oxidation reactions (Nat. Commun. 2019, 10, 234; Nat Commun 2020, 11, 335, ; J. Am. Chem. Soc. 2019, 141, 18, 7283).

Therefore, it is not quite clear what are the new insights/findings which the present work conveys.

Response: Thanks for your suggestion. Indeed, many single-atom catalysts (SACs) have been reported since the Pt₁/FeO_x was first reported for CO oxidation (Nat. Chem. 2011, 3, 634), such as Rh₁/γ-Al₂O₃ (ChemCatChem, 2013, 5, 1811), Rh₁/Co₃O₄ (ACS Catal. 2013, 3, 1011), Au₁/TiO₂ (J.Am.Chem.Soc. 2013, 135, 3768–3771) Au₁/CeO₂ (ACS Catal. 2015, 5, 6249), Au₁/FeO_x (Nano Res. 2015, 8, 2913), Pd₁/N-graphene (Adv. Mater. 2019, 1900509), Pd₁/La-

Al₂O₃ (Nat. Commun. 2014, 5, 4885), Pt₁/TiO₂ (Nat. Mater. 2019, 18, 746), Pt₁/FeO_x (Nat. Comm. 2019, 10, 234), Pt₁/Fe₂O₃ (Nat. Comm. 2019, 10, 4500), Pt₁/CeO₂ (Science, 2016, 353, 6295; Science, 2017, 358, 1419), M (Pt/Pd/Rh)/CeO₂-Al₂O₃ (Nat. Catal. 2020, 3, 368), Rh₁/ZrO₂ (J. Am. Chem. Soc. 2017, 139, 17694), Ir₁/FeO_x (J. Am. Chem. Soc. 2013, 135, 15314), Ce_{0.95}Ni_{0.025}Ru_{0.025}O₂ (J. Am. Chem. Soc. 2019, 141, 18, 7283) for several kinds of heterogeneous catalysis reactions. We presented at least three novelties in our work:

(1) Solvent-free synthesis: The reported synthesis method of SACs (including sintering-stable SACs) mainly focus on the wet-chemistry protocols, such as the co-precipitation method (Nat. Comm. 2019, 10, 234), the wet impregnation method (Science, 2017, 358, 1419;), the ion exchange method (J. Am. Chem. Soc. 2018, 140, 7407), the sonicated adsorption method (Nat Commun 2020, 11, 335, Nat. Catal. 2020, 3, 368) and the hydrothermal method (J. Am. Chem. Soc. 2019, 141, 7283). By comparison, the mechanochemical chemistry approach in our work provides a solid-state, fast, and efficient route for the preparation of SACs. It is no doubt that Pd-ZnO (Cell Reports Physical Science 2020, 1, 100004) showed the synthesis of SACs under solvent-free conditions. However, the used metal precursors were limited to acetylacetonate salt, and longer ball milling time (10 h) as well as the relatively lower calcination temperature (400 °C) were required to realize the formation of single Pd atoms. Additionally, our Pd₁@HEFO catalyst not only contains single Pd atoms but also a high surface area.

(2) Sintering-stable SACs from high configurational entropy: It is no doubt that Pt-FeO_x (Nat. Commun. 2019, 10, 234) and Pt-CeO₂ (Science 2016, 353, 150) can be synthesized under high-temperature treatment via a trapping method. This atom trapping protocol reported first from Datye and co-workers (Science 2016, 353, 150) is very useful for preparation of Pt single atoms at high temperature such as 800 °C because of the evaporation of PtO₂ and reducible CeO₂ or FeO_x that can continuously bind the mobile species. But this method wasn't extended to other metals like Pd due to the differences in chemical properties of different metals to the best of our knowledge. Additionally, single-atoms supported on metal oxides can afford the high-temperature thermal treatment and keep single-atom structures via some chemical methods such as grafting of N-rich organic linkers (Nat. Commun. 2020, 11, 335). In our work, the high configurational entropy plays an important role on the synthesis of SACs. In order to confirm the role of high configurational entropy in this work, we included Pd@ZrO₂, Pd@La₂O₃, Pd@HfO₂, Pd@TiO₂, ternary Pd@CeZrTiO_x and quaternary Pd@CeZrHfTiO_x samples synthesized by the

same method as Pd₁@HEFO and Pd@CeO₂. XRD patterns of all samples were displayed in Supplementary Fig. 8. The diffraction peaks at ~40.2° or ~42.0° ascribed to metallic Pd and PdO, respectively, can be clearly observed in single, ternary and quaternary metal oxides supported Pd catalysts. In sharp contrast, no diffraction peaks attributed to Pd species can be observed in Pd@CeZrHfTiLaO_x (Pd₁@HEFO) from Fig. 1, which is clear evidence that the formation of Pd SACs in Pd₁@HEFO originates from the high configurational entropy rather than the facile interactions between Pd and one of support elements.

Supplementary Fig. 8 PXRD of Pd@CeO₂, Pd@ZrO₂, Pd@La₂O₃, Pd@HfO₂, Pd@TiO₂, Pd@CeZrTiO_x and Pd@CeZrHfTiO_x samples.

(3) Outstanding thermal and hydrothermal stability: Although our Pd₁@HEFO is not the best catalyst and its CO oxidation activity is even worse than the Au catalyst (known as the most efficient active site for CO oxidation), it has outstanding thermal and hydrothermal stability. Due to the fact that pollution from vehicles' emissions usually includes CO, HC and NO_x, which requires high temperatures to treat three gaseous products simultaneously. Moreover, H₂O also co-exists in the exhausts, which often poison the catalysts. Therefore, it is very important to investigate the thermal and hydrothermal stability when a model reaction such as CO oxidation is carried out. Our Pd₁@HEFO catalyst in this work exhibited high low-temperature CO oxidation activity not only for the catalyst after calcination at 900 °C in air but also for that after

hydrothermal treatment at 750 °C for 10 h with 10 vol.% H₂O. The above harsh thermal and hydrothermal treatment conditions are closer to the severe reaction condition in practical applications. Additionally, we supplemented the oxidation activity of CO, C₃H₆ and NO over Pd₁@HEFO in Supplementary Figure 14a to expand the application scope of catalyst. Pd₁@HEFO exhibited very good oxidation activities of CO, C₃H₆ and NO at high gas hourly space velocity (GHSV) of 200,000 mL g cat⁻¹ hour⁻¹ with the total gas rate of 500 ml min⁻¹, although the T₁₀₀ of CO oxidation was shifted to ca. 260 °C due to the co-presence of C₃H₆ and NO and a high used GHSV. More importantly, the deactivation of CO, C₃H₆ and NO oxidation can be ignored over Pd₁@HEFO even after 10 hours of reaction in Supplementary Figure 14b, suggesting that Pd₁@HEFO showed a good activity and stability under a high GHSV.

All the above discussions suggest that entropy-stabilized Pd SACs on HEFO can be a candidate of catalyst for eliminating emissions from diesel engines. The related content has been supplemented as follows:

“More importantly, Pd₁@HEFO exhibits simultaneously outstanding oxidation activities of CO, C₃H₆ and NO at a high gas hourly space velocity (GHSV) of 200,000 mL gcat⁻¹ hour⁻¹ (Supplementary Fig. 14a), although the T₁₀₀ of CO oxidation shifts to ca. 260 °C due to the co-presence of C₃H₆ and NO and a high GHSV. The catalytic performance of Pd₁@HEFO is comparable to Pt/CeO₂-SiAlO_x regarded as a candidate of diesel oxidation catalyst (DOC)⁴⁸ and Pt/CeO₂⁴⁰. Moreover, no obvious deactivation of CO, C₃H₆, and NO oxidation can be observed over Pd₁@HEFO even after 10 hours of reaction in Supplementary Fig. 14b, implying that Pd₁@HEFO shows a good DOC activity and stability.” (Page 13 in the revised manuscript)

Supplementary Fig. 14 (a) CO, C₃H₆ and NO oxidation activity and (b) the stability of Pd₁@HEFO at 275 °C. Reaction conditions: 1000 ppm CO, 330 ppm C₃H₆, 200 ppm NO, 10%O₂, N₂ balance, a catalyst loading of 150 mg at a high gas hourly space velocity of 200,000 mL gcat⁻¹ hour⁻¹ with 500 ml min⁻¹ of the total gas flow rate.

Some other points:

1. In line 144, electron deficient Pd⁺⁴ is found for the catalyst by XPS, and later an intermediate oxidation between 0 and +2 (line 159) is found from XANES spectra. Why is that?

Response: The obtained binding energy (Supplementary Fig. 6) from XPS is the characteristic of electron-deficient Pd⁴⁺ (<4) in Pd₁@HEFO. However, XANES spectra show that the Pd K-edge absorption edge for Pd₁@HEFO is located between that of Pd foil and PdO (Fig. 3d), revealing the valence state of Pd is between 0 and +2. The difference between the valence state of Pd from XPS and XANES results come from the fact that XPS technology characterizes the surface elemental valence state while XANES as a more advanced technology can characterize the bulk elemental states and achieve the average valence of Pd in the bulk. Pd is easier to be oxidized to higher valence on the surface rather than on the bulk since surface Pd atoms are more likely to contact with oxygen, so the valence of surface Pd (XPS) is higher than that of bulk Pd (XANES). Therefore, the difference between the valences of Pd from XPS and XANES is reasonable and not contradictory. The related expressions have been revised as follows:

“XANES spectra show that the Pd K-edge absorption edge for Pd₁@HEFO is located between that of Pd foil and PdO (Fig. 3d), revealing the valence state of Pd is between 0 and +2, which is lower than +4 of the surface Pd from XPS. This is probably attributed to that the surface Pd atoms are more likely to contact with oxygen and be oxidized at high temperature.” (Page 8-9 in the revised manuscript)

2. The stability of the catalyst was tested, but low conversion was obtained (30%, Figure 4f inset). How was the stability at higher conversions?

Response: Thanks for your suggestions. The stability of Pd₁@HEFO at 170 °C with 100% CO conversion was supplemented and shown in the inset of Fig. 4f. The stability of 100% CO conversion over Pd₁@HEFO at 170 °C is similar to that of the catalyst at low CO conversion at 165 °C. CO conversion of Pd₁@HEFO remains stable in the first 50 hours, but a very slightly decreased trend is observed in the next 50 hours from the inset of Fig. 4f, finally, Pd₁@HEFO

still obtained ca. 98% CO conversion after 100 hours usage. The curve of the stability has been revised in the inset of Fig. 4f. (Page 14, in the revised manuscript).

Fig. 4f (inset) The stability of Pd₁@HEFO for CO oxidation reaction at 170 °C. Reaction conditions: A catalyst loading of 20 mg, 1 vol.% and CO balance in air at a gas hourly space velocity of 40,000 mL gcat⁻¹ hour⁻¹.

3. The deactivation test by water vapour was performed ex-situ, but for practical reasons water could be present during the catalytic reaction.

Response: The catalytic activity of CO oxidation in the presence of H₂O was also included and shown in Supplementary Figure 13, to investigate the effect of in-situ H₂O addition on the deactivation of CO oxidation over Pd₁@HEFO:

“In presence of 10 vol.% H₂O in the feed, the catalyst shows no deactivation from Supplementary Fig. 13. Instead, the activity of CO oxidation is slightly enhanced in the presence of 10 vol.% H₂O, especially in the low-temperature ranges, which is similar as that of Pd₁@HEFO-HA. It suggests that Pd₁@HEFO exhibited a good tolerance with H₂O.” (Page 14 in the revised Supplementary Information).

Supplementary Fig. 13 The effect of H₂O on the catalytic activity of CO oxidation over Pd₁@HEFO. Reaction conditions: A catalyst loading of 20 mg, 1 vol.% CO, 10 vol.% H₂O and balance in air at a gas hourly space velocity of 40,000 mL gcat⁻¹ hour⁻¹.

REVIEWERS' COMMENTS:

Reviewer #1 (Remarks to the Author):

I have read through the revised manuscript and “response to reviewers”. The authors have done many experiments according to the comments of the reviewers and some new data have been put into the revised manuscript and the support information. Most concerns have been addressed. Although many papers have been published for the SACs, fabricating SACs with thermodynamic stability on high-entropy supports by using a solvent-free synthetic strategy is very interesting. Therefore, I am happy to recommend this paper for publication in Nature Communications. Only several minor points should be considered before the publication:

Please check the procedure of CO adsorption (Page 19, line 365-372).

Please double check the related data in the revised manuscript and the data in the “response to reviewer”:

CO (11.10 $\mu\text{mol/pluse}$);

Adsorption amount 0.0644 $\mu\text{mol CO/}\mu\text{mol Pd}$ for Pd1@HEFO;

14.77 $\mu\text{L g}^{-1}$ of CO adsorbed on Pd1@HEFO sample

Page 5, line 101-102:

“EDS-mapping results shows the highly homogeneous dispersion of.....”

Should be “EDS-mapping results show the highly homogeneous dispersion of.....”?

Page 7, Line 122-123:

In the revised manuscript, authors have confirmed that only a small amount of Pd atoms are exposed (6%), but they still claimed that “...suggests that a small portion of Pd may be incorporated into HEFO sublattice for the formation of Pd_yCeZrHfTiLaOx solid solution”. It should be changed to “a large portion” instead of “a small portion”.

Page 8, line 175-176:

“Fig. 3 (a) HAADF-STEM image of Pd1@HEFO and corresponding EDS mapping of (b) Pd.”

Should be “Fig. 3 (a) HAADF-STEM image of Pd1@HEFO and (b) corresponding EDS mapping of Pd.”?

Page 15, line 277-278:

“A catalyst loading of 20 mg, 1 vol.% and CO balance in air”

Should be “A catalyst loading of 20 mg, and 1 vol.% CO balance in air”?

Two new references about solvent-free synthesis of SACs should be included in the revised manuscript. (Nature Communications 11, 1263 (2020); Cell Report Phys. Sci. 1, 10004 (2020).)

Reviewer #2 (Remarks to the Author):

The experimental work that the authors have conducted is certainly impressive. All the technical points have been addressed. The work merits publication, but it would require a minor revision in order to communicate better the key points.

This is because the previous comments regarding:

- i) "placing the work in perspective with respect to the current state-of-the-art", and
- ii) "the manuscript should be re-evaluated/re-written. In the current form, it is not apparent whether there is significant novelty or not for further consideration in the journal." have not been adequately addressed.

Since this work aims to a broad-interest journal, the key findings should/could be presented within a broader perspective and in a more consistent way.

In particular, the authors, replying to the previous comment from reviewer #2:

"Therefore, it is not quite clear what are the new insights/findings which the present work conveys. The attempts of the authors to put the work in perspective with the current state of the art, is probably limited on too general comparison/statements to properly highlight the novelty.", they mentioned three novelties: 1. Solvent-free synthesis, 2. Sintering-stable SACs from high configurational entropy; 3. Outstanding thermal and hydrothermal stability for CO oxidation. Regarding claim "2", authors explained in the reply that this refers particularly to Pd. If the authors found a way for sintering-stable Pd single-atom catalysts which was not achieved before, it is of very broad interest. Nevertheless, this is not presented in the key points of the work (i.e. abstract, introduction-rational or conclusions).

The same stands for the claim of outstanding thermal and hydrothermal stability – this information is missing in the abstract and Introduction part. Although, in the conclusions the authors mentioned:

"The as-synthesized single-atom Pd catalyst displays not only superior CO oxidation activity but also outstanding resistance to hydrothermal degradation compared to Pd@CeO₂ counterpart prepared using the same method.", this statement leaves the impression that the outstanding stability is attained in comparison only to some control materials presented in this work; in this way the broader importance/novelty is lost.

In another example, the current abstract mentions: "More importantly, the significantly improved reducibility of lattice oxygen existing in Pd-O-M form for Pd₁@HEFO are observed compared to Pd-O-Pd bonds of Pd@CeO₂ in CO atmosphere, thus exhibiting extremely higher low-temperature CO oxidation activity.", nevertheless, the authors mentioned in their reply to the revisions from reviewer #2, that the activity is not the best, but the stability at high temperatures and in presence of other gases (water vapours, NO and C₃H₆) is outstanding. Therefore, this is contradicting, pointing to the need for clarifying the key findings consistently and more clearly.

The rational which is developed in the introduction highlights only the technical novelty of preparing sintering-stable single atom catalysts without using a solvent. This is also interesting, but if this is the only new insight of the work (which seems it is not), then probably it is not of a broad interest.

As a result, the previous comments from reviewer #2 regarding:

- i) "placing the work in perspective with respect to the current state-of-the-art",.
- ii) "the manuscript should be re-evaluated/re-written. In the current form, it is not apparent whether there is significant novelty or not for further consideration in the journal."
could be addressed better.

In summary, from the authors reply and from the additional experiments that were presented it appears that the work merits publication, but the most important findings must be presented in a broad perspective and in a more consistent way providing a clear comparison with state-of-the-art systems.

Response to reviewers

REVIEWERS' COMMENTS:

Reviewer #1 (Remarks to the Author):

I have read through the revised manuscript and “response to reviewers”. The authors have done many experiments according to the comments of the reviewers and some new data have been put into the revised manuscript and the support information. Most concerns have been addressed. Although many papers have been published for the SACs, fabricating SACs with thermodynamic stability on high-entropy supports by using a solvent-free synthetic strategy is very interesting. Therefore, I am happy to recommend this paper for publication in Nature Communications. Only several minor points should be considered before the publication:

1. Please check the procedure of CO adsorption (Page 19, line 365-372). Please double check the related data in the revised manuscript and the data in the “response to reviewer”: CO (11.10 $\mu\text{mol/pluse}$); Adsorption amount 0.0644 $\mu\text{mol CO}/\mu\text{mol Pd}$ for Pd₁@HEFO; 14.77 $\mu\text{L g}^{-1}$ of CO adsorbed on Pd₁@HEFO sample.

Response: Thanks for your detailed comments. Following your valuable advice, we have carefully checked the data you mentioned above. It was found that the value of 14.77 μL was based on the 0.1074 g of Pd₁@HEFO sample. Therefore, the actual amount of CO adsorbed on Pd₁@HEFO sample should be $14.77/0.1074 \mu\text{L g}^{-1} = 137.52 \mu\text{L g}_{\text{cat}}^{-1}$, and the corresponding CO adsorption still is 0.0644 $\mu\text{mol CO}/\mu\text{mol Pd}$. To avoid any confusion, “The absence of Pd atoms outside HEFO lattice (Fig. 4a) and around 6.44% surface Pd atoms determined by CO chemisorption (0.0644 $\mu\text{mol CO}/\mu\text{mol Pd}$, Supplementary Table 1).....” was added in the revised manuscript (on Page 9).

2. Page 5, line 101-102:

“EDS-mapping results shows the highly homogeneous dispersion of.....” Should be “EDS-mapping results show the highly homogeneous dispersion of.....”?

Page 7, Line 122-123:

In the revised manuscript, authors have confirmed that only a small amount of Pd atoms are exposed (6%), but they still claimed that "...suggests that a small portion of Pd may be incorporated into HEFO sublattice for the formation of Pd_yCeZrHfTiLaO_x solid solution". It should be changed to "a large portion" instead of "a small portion".

Page 8, line 175-176:

"Fig. 3 (a) HAADF-STEM image of Pd₁@HEFO and corresponding EDS mapping of (b) Pd." Should be "Fig. 3 (a) HAADF-STEM image of Pd₁@HEFO and (b) corresponding EDS mapping of Pd."?

Page 15, line 277-278:

"A catalyst loading of 20 mg, 1 vol.% and CO balance in air" Should be "A catalyst loading of 20 mg, and 1 vol.% CO balance in air"?

Response: Thanks for your valuable recommendations. Based on your advice, we have corrected the above-mentioned mistakes in the revised manuscript. In addition, we have thoroughly reviewed the manuscript and have carefully corrected the grammatical errors. The detailed revisions have been highlighted in the revised manuscript.

"EDS-mapping results shows the highly homogeneous dispersion of....." was revised to "EDS-mapping results show the highly homogeneous dispersion of....."?

"a small portion" was revised to "a large portion".

"Fig. 3 (a) HAADF-STEM image of Pd₁@HEFO and corresponding EDS mapping of (b) Pd." was revised to "Fig. 4 (a) HAADF-STEM image of Pd₁@HEFO and (b) corresponding EDS mapping of Pd."?

"A catalyst loading of 20 mg, 1 vol.% and CO balance in air" was revised to "A catalyst loading of 20 mg, and 1 vol.% CO balance in air".

"Mechanochemistry scenarioshowever, has always been a great challenge for fabricating atomically dispersed metal sites." was revised to "Mechanochemistry scenarioshowever, have always been a great challenge for fabricating atomically dispersed metal sites." (Page 3 in the revised manuscript)

“Fortunately, the surface area, pore size distribution, and crystalline structure of Pd₁@HEFO keeps almost unchanged after both thermal and hydrothermal treatment.” Was revised to “Fortunately, the surface area, pore size distribution, and crystalline structure of Pd₁@HEFO stay almost unchanged after both thermal and hydrothermal treatment.” (Page 8 in the revised manuscript)

.....

3. Two new references about solvent-free synthesis of SACs should be included in the revised manuscript. (Nature Communications 11, 1263 (2020); Cell Report Phys. Sci. 1, 10004 (2020).)

Response: Thanks for your suggestions. The related references have been updated in the revised manuscript:

17. He, X. *et al.* Mechanochemical kilogram-scale synthesis of noble metal single-atom catalysts, *Cell Report Phys. Sci.* **1**, 10004 (2020).

18. Liu, K. *et al.* Strong metal-support interaction promoted scalable production of thermally stable single-atom catalysts, *Nat. Commun.* **11**, 1263 (2020).

Reviewer #2 (Remarks to the Author):

The experimental work that the authors have conducted is certainly impressive. All the technical points have been addressed. The work merits publication, but it would require a minor revision in order to communicate better the key points.

This is because the previous comments regarding:

- i) “placing the work in perspective with respect to the current state-of-the-art”, and
- ii) “the manuscript should be re-evaluated/re-written. In the current form, it is not apparent whether there is significant novelty or not for further consideration in the journal.” have not been adequately addressed.

Since this work aims to a broad-interest journal, the key findings should/could be presented within a broader perspective and in a more consistent way.

In particular, the authors, replying to the previous comment from reviewer #2: “Therefore, it is not quite clear what are the new insights/findings which the present work conveys. The attempts of the authors to put the work in perspective with the current state of the art, is probably limited on too general comparison/statements to properly highlight the novelty.”, they mentioned three novelties: 1. Solvent-free synthesis, 2. Sintering-stable SACs from high configurational entropy; 3. Outstanding thermal and hydrothermal stability for CO oxidation. Regarding claim “2”, authors explained in the reply that this refers particularly to Pd. If the authors found a way for sintering-stable Pd single-atom catalysts which was not achieved before, it is of very broad interest. Nevertheless, this is not presented in the key points of the work (i.e. abstract, introduction-rational or conclusions).

Response: Thanks for your suggestion. According to reviewer’s comment, the manuscript was re-written to highlight the novelty of this work, which we reported a possibility of Pd₁ stabilized on metal oxides at a given temperature with pronounced entropy effect to solve the aggregation issue of Pd₁. Therefore, the synthesis of sintering-resistant Pd₁ is the major novelty in our work, during that the entropy-stabilized protocol is considered as the means.

We re-wrote the manuscript, especially the abstract portion, to help readers better capture the major novelty of this manuscript and attract broader interest as shown below:

Abstract: Single-atom catalysts (SACs) have attracted considerable attention in the catalysis community. However, fabricating intrinsically stable SACs on traditional supports (N-doped carbon, metal oxides, etc.) remains a formidable challenge, especially under high-temperature conditions. Here, we report a novel entropy-driven strategy to stabilize Pd single-atom on the high-entropy fluorite oxides (CeZrHfTiLa)O_x (HEFO) as the support by a combination of mechanical milling with calcination at 900 °C. Characterization results reveal that single Pd atoms are incorporated into HEFO (Pd₁@HEFO) sublattice by forming stable Pd–O–M bonds (M=Ce/Zr/La). Compared to the traditional support stabilized catalysts such as Pd@CeO₂, Pd₁@HEFO affords the improved reducibility of lattice oxygen and the existence of stable Pd–O–M species, thus exhibiting not only higher low-temperature CO oxidation activity but also outstanding resistance to thermal and hydrothermal degradation. This work therefore exemplifies the superiority of high-entropy materials for the preparation of SACs.

Introduction

Therefore, fabricating sintering-resistant SACs with intrinsically thermodynamic stability on high-entropy supports by using a solvent-free synthetic strategy is highly desirable.

The catalytic activity of CO oxidation, as well as the resistance to thermal and hydrothermal degradation are then compared for Pd₁@HEFO and Pd@CeO₂ catalysts to prove the advantages of HEFO as the catalyst carrier.

Result

As is well known, the phase stabilization is a process determined by combination of the enthalpy (H) and entropy (S) effects, which are temperature- and composition-dependent. Compared to the Pd@CeO₂, Pd₁@HEFO with enhanced compositional complexity provides a higher molar configurational entropy, especially for equimolar cations, which then potentially decreases the Gibbs free energy according to the equation ($\Delta G = \Delta H - T\Delta S$). This means that the formation of (Pd_yCeZrHfTiLa)_x solid solution is an entropy-dominated process, whereas the decreased configuration entropy induces the dissociation of (Pd_yCe)_x as an enthalpy-driven process. To prove this hypothesis, the PXRD patterns of Pd@ZrO₂, Pd@La₂O₃, Pd@HfO₂, Pd@TiO₂, ternary Pd@CeZrTiO_x, and quaternary Pd@CeZrHfTiO_x samples synthesized by the same method are also collected. The diffraction peaks ascribed to metallic Pd and/or PdO can be observed, further confirming the importance of the high configurational entropy on stabilizing the Pd single atoms. We also synthesized the Pd/HEFO-p (single Pd atoms dispersed on HEFO carrier) by a post-deposition method, where the single atom structure can be confirmed by XRD pattern and CO-DRIFTS spectra in Supplementary Fig. 9. Unfortunately, the sintering and aggregation of Pd on HEFO can be obviously observed in Pd/HEFO-p-900 after post-treatment at 900 °C, which might be ascribed to the excessive Pd density on HEFO surface. (on Page 12)

Discussion

In summary, we have developed a solid-state strategy to synthesize a sintering-resistant Pd single-atom catalyst stabilized on HEFO (Pd₁@HEFO). The as-synthesized single-atom Pd catalyst displays not only superior CO oxidation activity but also outstanding resistance to thermal and hydrothermal degradation compared to Pd@CeO₂ counterpart prepared using the same method. The choice of host in this work plays a paramount role on the synthesis of single-atom Pd catalysts, which can only be realized with HEFO as the carrier because of its maximum configurational entropy. This trait induces Pd to be incorporated into the HEFO sublattice during

the mechanochemical-assisted preparation process, and the above process cannot be accomplished with CeO₂ as an alternative carrier. This work provides a solvent-free entropy-driven methodology for the synthesis of SACs and may guide the development of next-generation SACs.

The same stands for the claim of outstanding thermal and hydrothermal stability – this information is missing in the abstract and Introduction part. Although, in the conclusions the authors mentioned: “The as-synthesized single-atom Pd catalyst displays not only superior CO oxidation activity but also outstanding resistance to hydrothermal degradation compared to Pd@CeO₂ counterpart prepared using the same method.”, this statement leaves the impression that the outstanding stability is attained in comparison only to some control materials presented in this work; in this way the broader importance/novelty is lost.

Response: We included the description of thermal and hydrothermal stability in abstract and introduction parts as suggested. Moreover, we also supplemented the thermal and hydrothermal stability of other catalysts (such as Pd/CeO₂, *ACS Catal.* 2017, 7, 6887; Pd/TiO₂, *Prog. Nat. Sci.* 2016, 26, 289; Au/CeO₂, *ACS Catal.* 2015, 5, 1653; Pt/CeO₂, *Science* 2017, 358, 1419; etc.) during CO oxidation in addition to the Pd@CeO₂ and Pd@HEFO in our work. We added one sentence in the revised manuscript to clarify these comparisons: “Additionally, Pd₁@HEFO not only exhibits better thermal and hydrothermal stability than Pd@CeO₂ in this work, but also shows better or comparable performance relative to other reported representative catalysts of CO oxidation¹⁻⁴.” (Page 14 in the revised manuscript)

In another example, the current abstract mentions: “More importantly, the significantly improved reducibility of lattice oxygen existing in Pd-O-M form for Pd₁@HEFO are observed compared to Pd-O-Pd bonds of Pd@CeO₂ in CO atmosphere, thus exhibiting extremely higher low-temperature CO oxidation activity.”, nevertheless, the authors mentioned in their reply to the revisions from reviewer #2, that the activity is not the best, but the stability at high temperatures and in presence of other gases (water vapours, NO and C₃H₆) is outstanding. Therefore, this is contradicting, pointing to the need for clarifying the key findings consistently and more clearly. The rationale which is developed in the introduction highlights only the technical novelty of

preparing sintering-stable single atom catalysts without using a solvent. This is also interesting, but if this is the only new insight of the work (which seems it is not), then probably it is not of a broad interest.

Response: In the response to Reviewer # 2, we added “Although our Pd₁@HEFO is not the best catalyst in terms of its CO oxidation activity, it has outstanding thermal and hydrothermal stability.” We do not claim that our catalyst has the highest catalytic activity.

To avoid any confusion, “More importantly, the significantly improved reducibility of lattice oxygen existing in Pd-O-M form for Pd₁@HEFO are observed compared to Pd-O-Pd bonds of Pd@CeO₂ in CO atmosphere, thus exhibiting extremely higher low-temperature CO oxidation activity.” has been revised to “Compared to the traditional support-stabilized catalysts such as Pd@CeO₂, Pd₁@HEFO affords the improved reducibility of lattice oxygen and the existence of stable Pd–O–M species, thus exhibiting not only higher low-temperature CO oxidation activity but also outstanding resistance to thermal and hydrothermal degradation.” in the Abstract.

As a result, the previous comments from reviewer #2 regarding:

- i) “placing the work in perspective with respect to the current state-of-the-art”,
 - ii) “the manuscript should be re-evaluated/re-written. In the current form, it is not apparent whether there is significant novelty or not for further consideration in the journal.”
- could be addressed better.

In summary, from the authors reply and from the additional experiments that were presented it appears that the work merits publication, but the most important findings must be presented in a broad perspective and in a more consistent way providing a clear comparison with state-of-the-art systems.

Response: Thanks for your suggestions. We have revised the related discussions in the revised manuscript to present the most important findings in a broad perspective and in a more consistent way providing a clear comparison with state-of-the-art systems:

Abstract: Single-atom catalysts (SACs) have attracted considerable attention in the catalysis community. However, fabricating intrinsically stable SACs on traditional supports (N-doped

carbon, metal oxides, etc.) remains a formidable challenge, especially under high-temperature conditions. Here, we report a novel entropy-driven strategy to stabilize Pd single-atom on the high-entropy fluorite oxides (CeZrHfTiLa) O_x (HEFO) as the support by a combination of mechanical milling with calcination at 900 °C. Characterization results reveal that single Pd atoms are incorporated into HEFO (Pd₁@HEFO) sublattice by forming stable Pd–O–M bonds (M=Ce/Zr/La). Compared to the traditional support stabilized catalysts such as Pd@CeO₂, Pd₁@HEFO affords the improved reducibility of lattice oxygen and the existence of stable Pd–O–M species, thus exhibiting not only higher low-temperature CO oxidation activity but also outstanding resistance to thermal and hydrothermal degradation. This work therefore exemplifies the superiority of high-entropy materials for the preparation of SACs.

Introduction

Therefore, fabricating sintering-resistant SACs with intrinsically thermodynamic stability on high-entropy supports by using a solvent-free synthetic strategy is highly desirable.

The catalytic activity of CO oxidation, as well as the resistance to thermal and hydrothermal degradation are then compared for Pd₁@HEFO and Pd@CeO₂ catalysts to prove the advantages of HEFO as the catalyst carrier.

Result

As is well known, the phase stabilization is a process determined by combination of the enthalpy (H) and entropy (S) effects, which are temperature- and composition-dependent. Compared to the Pd@CeO₂, Pd₁@HEFO with enhanced compositional complexity provides a higher molar configurational entropy, especially for equimolar cations, which then potentially decreases the Gibbs free energy according to the equation ($\Delta G = \Delta H - T\Delta S$). This means that the formation of (Pd_yCeZrHfTiLa) O_x solid solution is an entropy-dominated process, whereas the decreased configuration entropy induces the dissociation of (Pd_yCe) O_x as an enthalpy-driven process. To prove this hypothesis, the PXRD patterns of Pd@ZrO₂, Pd@La₂O₃, Pd@HfO₂, Pd@TiO₂, ternary Pd@CeZrTiO_x, and quaternary Pd@CeZrHfTiO_x samples synthesized by the same method are also collected. The diffraction peaks ascribed to metallic Pd and/or PdO can be observed, further confirming the importance of the high configurational entropy on stabilizing the Pd single atoms. We also synthesized the Pd/HEFO-p (single Pd atoms dispersed on HEFO carrier) by a post-deposition method, where the single atom structure can be confirmed by XRD

pattern and CO-DRIFTS spectra in Supplementary Fig. 9. Unfortunately, the sintering and aggregation of Pd on HEFO can be obviously observed in Pd/HEFO-p-900 after post-treatment at 900 °C, which might be ascribed to the excessive Pd density on HEFO surface. (on Page 12)

Additionally, Pd₁@HEFO not only exhibits better thermal and hydrothermal stability than Pd@CeO₂ in this work, but also shows better or comparable performance relative to other reported representative catalysts of CO oxidation¹⁻⁴. (on Page 14)

Discussion

In summary, we have developed a solid-state strategy to synthesize a sintering-resistant Pd single-atom catalyst stabilized on HEFO (Pd₁@HEFO). The as-synthesized single-atom Pd catalyst displays not only superior CO oxidation activity but also outstanding resistance to thermal and hydrothermal degradation compared to Pd@CeO₂ counterpart prepared using the same method. The choice of host in this work plays a paramount role on the synthesis of single-atom Pd catalysts, which can only be realized with HEFO as the carrier because of its maximum configurational entropy. This trait induces Pd to be incorporated into the HEFO sublattice during the mechanochemical-assisted preparation process, and the above process cannot be accomplished with CeO₂ as an alternative carrier. This work provides a solvent-free entropy-driven methodology for the synthesis of SACs and may guide the development of next-generation SACs.

References

1. Spezzati, G. *et al.* Atomically dispersed Pd–O Species on CeO₂ (111) as highly active sites for low-temperature CO oxidation, *ACS Catal.* **7**, 6887–6891 (2017).
2. Nie, L. *et al.* Activation of surface lattice oxygen in single-atom Pt/CeO₂ for low-temperature CO oxidation, *Science* **358**, 1419–1423 (2017).
3. Bai, Y., Wang, C., Zhou, X., Lu, J. & Xiong, Y. Atomic layer deposition on Pd nanocrystals for forming Pd-TiO₂ interface toward enhanced CO oxidation, *Prog. Nat. Sci.* **26**, 289-294 (2016).
4. Slavinskaya, E. M. *et al.* Low-temperature CO oxidation by Pd/CeO₂ catalysts synthesized using the coprecipitation method, *Appl. Catal. B: Environ.* **166-167**, 91-103 (2015).